# Dancing with Discrepancies: Commonality-Specificity Attention GAN for Weakly Supervised Medical Lesion Segmentation

## Abstract

Increasing weakly supervised semantic segmentation methods concentrate on the target segmentation by leveraging solely image-level labels. However, few works notice that a significant gap exists in addressing medical characteristics, which demands massive attention. In this paper, we note: (i) Lesion regions typically exhibit a sharp probability distribution pattern while healthy tissues adhere to an underlying homogeneous distribution, which deviates from typical natural images; (ii) Boundaries of lesion foregrounds and structural backgrounds are blurred; (iii) Similar structures frequently appear within specific organs or tissues, which poses a challenge to concentrating models' attention on regions of interest instead of the entire image. Thus we propose a **Co**mmonality-spec**i**ficity atte**n**tion **GAN** (**CoinGAN**) to overcome the above challenges, which leverages distribution discrepancies to mine the knowledge underlying images. Specifically, we propose a new form of convolution, *contrastive convolution*, to utilize the fine-grained perceptual discrepancies of activation sub-maps to enhance the intra-image distribution, making lesion foregrounds (specificity) and structural backgrounds (commonality) boundary-aware. Then a *commonality-specificity attention mechanism* and the GAN-based loss function are devised to jointly suppress similarity regions between different labels of images and accentuate discrepancy regions between different labels of images. This isolates lesion areas from the structural background. Extensive experiments are conducted on three public benchmarks. Our CoinGAN achieves state-of-the-art performance with the DSC of 71.69%, 84.73%, and 78.32% on QaTa-COV19, ISIC2018, and MoNuSeg datasets, making a significant contribution to the detection of pneumonia, skin disease, and cancer. Furthermore, the visualized results also corroborate the effectiveness of CoinGAN in segmenting medical objects.

## 1 Introduction

Semantic segmentation has shown substantial progress in a diverse array of computer vision tasks, e.g., autonomous driving, robotics and medical diagnosis Mo et al. (2022). However, these models are heavily dependent on pixel-level annotations, which are notoriously laborious and time-intensive. On the contrary, some weak supervision alternatives, e.g., image-level labels He et al. (2024), points Gao et al. (2024), and bounding boxes Cheng et al. (2023), are easier to obtain. Therefore, exploring the potential of weak annotations for semantic segmentation is appealing. In this paper, we aim to advance weakly supervised semantic segmentation (WSSS) for medical images, utilizing solely the image-level annotations for supervision.

Image-level WSSS is extremely challenging since these image-level labels solely indicate the presence or absence of the target object without specifying any location information. To counter this, a pioneering approach, class activation maps (CAM) Zhou et al. (2016) endue convolutional neural networks with locating ability for recognizing the most discriminative regions. However, these location maps inevitably suffer from sparsity (false negative) or inappropriately activate false background structures for the target objects (false positive) Chen et al. (2022a;b). Such incomplete correspondence between location maps and actual object locations severely hampers the performance of CAM-like methods Lin et al. (2023b); Kim et al. (2024). Recent endeavors have spearheaded

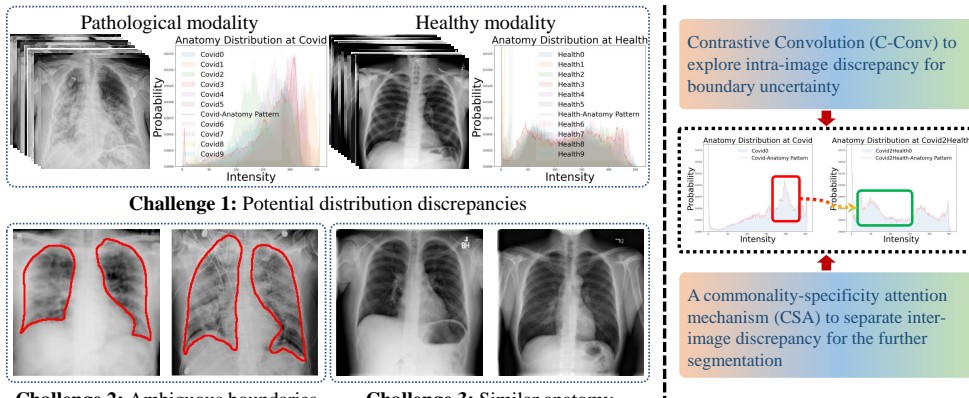

Figure 1: **Main challenges of medical images (left) and the core idea of our proposal (right).** Left: a visualized example of medical challenges. 1) Lesion regions tend to exhibit a sharp probability distribution pattern while healthy tissues adhere to an underlying homogeneous probability distribution pattern. 2) Boundaries of lesion foreground and structural background are ambiguous. 3) Similar anatomy structures are observed. Right: the core idea of our proposal. The above challenges are addressed by leveraging the distribution discrepancies between different labels of medical images, where the intra-image discrepancies are explored by the C-Conv module and the inter-image discrepancies are learned by the CSA mechanism.

initiatives to overcome this issue by introducing new network architectures, e.g., ViT Hanna et al. (2023), SAM Kweon & Yoon (2024), or new training strategies, including text-driven strategies Lin et al. (2023b), shared feature strategies Zhao et al. (2024b), but such models do not account for the causes of oversegmentation and inaccurate shapes[1] in medical segmentation.

**Our insight** is that, in medical WSSS, image-level labels are insufficient to address the key challenges inherent in medical images. For illustration, we take the coronavirus disease 2019 (COVID-19) as a visualized example (Figure 1 (left)): **i)** We plot the probability mass function (PMF) for two distinct labels in the medical images, where medical images belonging to the pathological modalities exhibit a sharp probability distribution pattern while those holding the healthy modalities adhere to an underlying homogeneous probability distribution pattern. Nonetheless, such pronounced distribution discrepancies render models vulnerable to pathological modalities, facilitating a hasty convergence through this 'shortcut'. This hinders the models' ability to thoroughly explore the discriminative regions. **ii)** Boundaries of lesion foreground and structural background are ambiguous. This phenomenon is widespread in lesion segmentation tasks. For instance, certain lesion tissues, e.g., ground-glass opacities of COVID-19 or brain glioblastoma, grow gradually alongside lungs or brains, making it challenging for models to delineate lesion boundaries. **iii)** Similar structures remain consistent in specific organs or neighboring tissues, which challenges the model's ability to focus on local representations rather than the entire image. The interference from similar structures further hinders the model's ability to explore complete discriminative regions.

Therefore, we propose a **Co**mmonality-spec**i**ficity atte**n**tion **GAN (CoinGAN)**, an innovative model to leverage the inherent discrepancies within regions of the same image and between different labels of images for enhancing the segmentation of objects. As in Figure 1(Right), the C-Conv module is devised to explore the intra-image distribution discrepancies, where fine-grained perceptual discrepancies of activation sub-maps within the same image are used to adaptively reweight the boundary representations for ensuring a clear distinction of different regions within images, which eliminates ambiguous boundaries. Subsequently, we propose the commonality-specificity attention (CSA) mechanism to recognize inter-image distribution discrepancies. In this process, similarity representations between different labels of images are suppressed and discrepancy representations between different labels of images are accentuated. This steers the model's attention toward the concerned lesion regions, eliminating the interference of similar structures. Finally, representations enhanced by the C-Conv module and CSA mechanism are fed into a GAN network and an adversarial loss function is used to drive the distribution conversion between different labels of images for enhancing the segmentation of objects. The three components above collaboratively eliminate the

---

[1]The concepts of oversegmentation and inaccurate shapes are provided in Appendix A.

issue of incomplete exploration caused by significant distribution differences. Our contributions are summarized as follows:

- We propose an innovative model, CoinGAN, to address the medical WSSS issue. It leverages distribution discrepancies underlying different labels of medical images to generate high-quality pixel-level annotations. To our best knowledge, CoinGAN is the first work to fully explore the distribution discrepancies in medical images for medical WSSS without any auxiliary information.

- An innovative convolution (C-Conv) and a dual attention (CSA) mechanism are proposed to explore latent distribution discrepancies. C-Conv concentrates on intra-image discrepancy learning to reduce boundary ambiguity. The CSA mechanism accentuates inter-image discrepancy learning for eliminating the interference of similar structures. The GAN-based adversarial loss function conducts the distribution conversion between different labels of medical images for complete object segmentation. The three components above collectively eliminate the interference from significant distribution differences.

- CoinGAN achieves state-of-the-art performance across three public benchmarks and the visualized results corroborate the effectiveness of the distribution conversion.

## 2 RELATED WORK

**Weakly Supervised Semantic Segmentation.** WSSS has attracted increasing attention via weak supervision alternatives, such as image-level annotations He et al. (2024), points Gao et al. (2024), and bounding boxes Cheng et al. (2023), which substantially alleviates the manual annotation burden. Among them, image-level annotations stand out for their minimal annotation costs, but the lack of location information presents a significant challenge. A mainstream solution, CAM Zhou et al. (2016), adds a global average pooling layer to the convolutional neural networks for generating location maps. However, these location maps usually highlight the most discriminative areas of the target or co-occurring objects, resulting in degraded segmentation performance. A plethora of research efforts have been proposed to refine these location regions, e.g., semantic association Zhang et al. (2020), boundary constraint Rong et al. (2023), threshold operation Lee et al. (2022b), auxiliary information Xu et al. (2021); Lee et al. (2022a), new training strategies Lin et al. (2023b); Zhao et al. (2024b) or network architectures Kweon & Yoon (2024). However, these methods suffer from a severe performance gap when directly applied to medical images owing to more challenging medical characteristics Chen et al. (2022a).

**Anatomical Priors.** Prior knowledge is extremely crucial in semantic segmentation tasks, which guides training models to converge in the correct direction Redondo-Cabrera et al. (2019). For instance, SkelCon Tan et al. (2022) leverages skeletal priors for retinal vessel segmentation. PaNN Zhou et al. (2019) incorporates abdominal anatomy priors into multi-organ segmentation. Min-Max Belharbi et al. (2021) utilizes the intra-category histology variations to segment colon cancer. Swin-MIL Qian et al. (2022) reveals inter-instance correction for colon cancer segmentation. SAM-driven Zhao et al. (2024a) depends on the pre-trained priors for the nodule segmentation. However, these methods are mainly designed for some specific organs or scenarios, and sometimes need additional auxiliary information. This considerably restricts the model's applicability. In contrast, our proposed CoinGAN aims to explore prior knowledge directly from medical images themselves without extra auxiliary information or tools.

**Attention Mechanisms.** The attention mechanism is playing an increasingly important role in computer vision tasks Rahman et al. (2024). It can direct attention to the most salient regions, providing a discriminative insight that holds significant potential in WSSS tasks. Recently, a few works have attempted attention mechanisms to WSSS. For instance, Group Zhou et al. (2022) proposes a co-attention mechanism to discover the semantic relations in images. MCTformer Xu et al. (2022) employs the multi-head self-attention mechanism to capture global class-specific attention. Sparse-VIT Hanna et al. (2023) inserts gating units to the multi-head attention mechanism for correlated region sparsity control. However, to our best knowledge, there are few similar attempts in medical WSSS. Inspired by it, our CoinGAN devises a commonality-specificity attention mechanism for medical WSSS, which integrates the distribution discrepancies from the image themselves and attention mechanisms to investigate the knowledge underlying different labels of images.

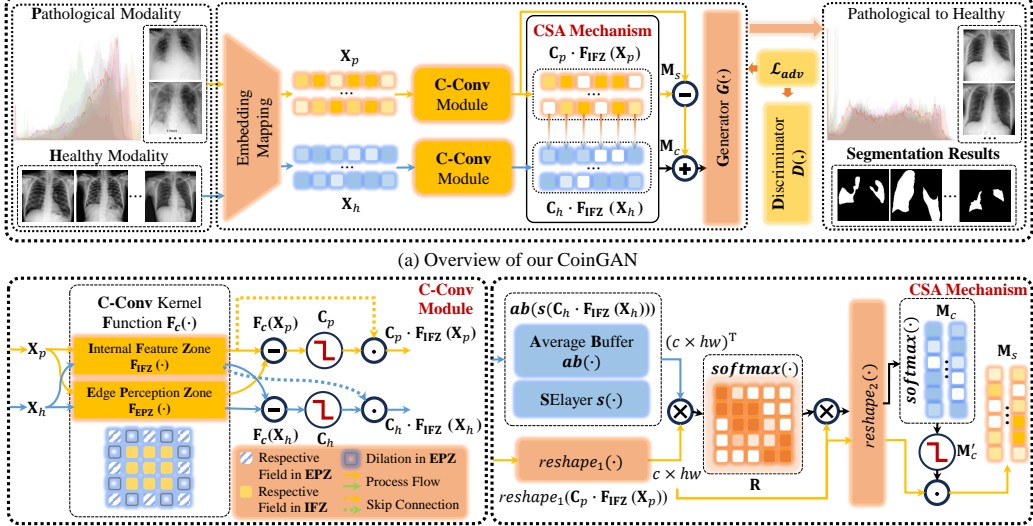

(a) Overview of our CoinGAN

(b) C-Conv Module

(c) CSA Mechanism

Figure 2: **CoinGAN targets to adaptively learn the representation distribution discrepancies to capture the latent knowledge gap inherent in the images themselves**, which contains three crucial components. The C-Conv module is devised to learn intra-image distribution discrepancies, reducing the boundary uncertainty. The CSA mechanism is proposed for the inter-image discrepancy learning. The resulting commonality map $M_c$ is added to intra-image reinforced features to enhance the background distribution while the specificity map $M_s$ filters these features to drive abnormal feature distributions to different labels of distributions. Finally, a GAN-based backbone facilitates the distribution conversion and further separates lesion regions.

## 3 METHODOLOGY

### 3.1 MOTIVATION & OVERVIEW

We analyze medical WSSS by two important questions. *Question 1:* Why do classification results tend to be accurate but the segmentation performance is not satisfactory? *Question 2:* Why does the segmentation shape severely deviate from ground truth contours? The first answer is that the classification model can establish associations between image-wise labels and discriminative regions. These discriminative regions have a strong statistical correlation with the aligned category, that is, the classification model discovers the category specificity, which is sufficient to make an accurate classification prediction (a 'shortcut' convergence). However, previous works Zhao et al. (2024b) have shown these specific regions also present a strong association with surrounding regions, this is because the boundary ambiguity leads to an extended segmentation region, and further causes the oversegmentation of objects. For instance, when segmenting pneumonia lesions, we may get the entire thoracic cavity. The second answer is that the output results lack the constraints of background structures, failing to prevent the model's attention from deviating from object regions. This is mainly because pronounced distribution discrepancies between labels, as well as similar structures within the same image label, further exacerbate the phenomenon of "shortcut" convergence, which makes the model fail to effectively capture fine-grained local representations, resulting in inaccurate segmentation shapes.

Therefore, we propose the **C-Conv** module and **CSA** mechanism. Specifically, the C-Conv module is proposed to learn the intra-image distribution discrepancies for reducing the ambiguous boundaries, which forms a clear distinction of different regional distributions within the image and prevents oversegmentation. Subsequently, the CSA mechanism is designed to learn the inter-image distribution discrepancies for mitigating the inference of similar structures, which captures the similarity regions and discrepancy regions between different labels of images to help enhance the shape constraints. Finally, for the pronounced distribution discrepancies across different labels, a GAN-based adversarial loss function assists the model in performing the global distribution conversion and further extracting valuable regions of interest, thereby forming the precise segmentation. The overview of CoinGAN is illustrated in Figure 2.

## 3.2 CONTRASTIVE CONVOLUTION (C-CONV) MODULE

In medical WSSS tasks, different labels of images $\mathbf{I}$ are first projected into the embedding space by an Embedding Mapping layer $\mathbf{EM}(\cdot)$, thereby obtaining initial representations $\mathbf{X}$:

$$\mathbf{X} = \mathbf{EM}(\mathbf{I}) \tag{1}$$

where the initial representations $\mathbf{X} = \{x_{u,v} | 0 \le u < h, 0 \le v < w\}$ ($h, w$ are the height and width of representations $\mathbf{X}$) are rich in image details and generally fed into deeper neural networks to extract high-level representations. In this process, convolution plays a pivotal role[2]:

$$
\begin{aligned}
\mathbf{F}(x_{u,v}) &= \sum_{p,q}^{k,k} \mathbf{W}_{p,q} x_{u-\lfloor k/2 \rfloor + p, v - \lfloor k/2 \rfloor + q} \\
&= \sum_{p,q}^{k,k} \mathbf{W}_{p,q} x_{u + \triangle u, v + \triangle v}
\end{aligned}
\tag{2}
$$

where for the input representation $x_{u,v}$, its output representation $\mathbf{F}(x_{u,v})$ (a standard convolution) is computed within a receptive field of $k \times k$ size. The element-wise multiplication and summation are conducted using the kernel weights $\mathbf{W} \in \mathbb{R}^{k \times k}$ at the kernel position $(p, q)$ and the corresponding position of pixel $x$. $\triangle u$ and $\triangle v$ denote the corresponding positional offsets.

Despite achieving great success in learning local representations, such convolutional structures may introduce certain irrelevant positions of weighted representation $x^{ir}$ into the representations $\mathbf{F}(x_{u,v})$ as a noise term. This is because the boundary regions are typically situated at the intersection of the target foreground and structural background; hence, their element-wise weighted average inevitably expands and blurs the boundaries. This further causes discriminative object information to leak into background regions, that is, the cause of oversegmentation as elaborated in *Question 1*.

Formally, these mixed-weighted representation corresponding to target regions can be defined as:

$$\mathbf{F}(x_{u,v}) = \sum_{p,q \in \mathcal{R}} \mathbf{W}_{p,q} x^r_{u + \triangle u, v + \triangle v} + \sum_{p,q \in \mathcal{IR}} \mathbf{W}_{p,q} x^{ir}_{u + \triangle u, v + \triangle v} \tag{3}$$

where the relevant representations $x^r_{u + \triangle u, v + \triangle v}$ contribute positively to discernible regions ($p, q \in \mathcal{R}$), while the irrelevant representations $x^{ir}_{u + \triangle u, v + \triangle v}$ negatively influence the model's decision-making by introducing confusion ($p, q \in \mathcal{IR}$).

Thus we propose a new form of convolution, C-Conv, to learn intra-image distribution discrepancies for addressing the above ambiguous representations. As depicted in Figure 2(b), the C-Conv kernel consists of an *Edge Perception Zone (EPZ)* and an *Internal Feature Zone (IFZ)*. EPZ has a wider receptive field, enabling it to identify the potential change regions earlier, while IFZ focuses more on capturing local representations within a small range. Therefore, an intuitive idea comes out. For the same position of representation, the representation discrepancy learned by EPZ and IFZ will be a crucial criterion for determining whether the representation is ambiguous. For instance, for the same category of areas, the corresponding distribution within an $n \times n$ receptive field tends to be similar. Thus, the convolutional discrepancy between EPZ and IFZ is small, hovering around 0. However, when a boundary emerges, the EPZ convolution, with its wider receptive field, will undergo changes earlier and detect the presence of a new category. In contrast, the IPZ convolution within a small receptive field remains relatively stable. Therefore, it is promising to learn intra-image distributions by exploiting the fine-grained perceptual discrepancy between both types of convolutions. Specifically, C-Conv $\mathbf{F}_c(.)$ can be formulated as:

$$\mathbf{F}_c(x_{u,v}) = \sum_{s,t}^{k',k'} \mathbf{W}_{s,t} x_{u-k'+1+2s, v-k'+1+2t} - \mathbf{F}(x_{u,v}) \tag{4}$$

$$= \mathbf{F}_{\text{EPZ}}(x_{u,v}) - \mathbf{F}_{\text{IFZ}}(x_{u,v})(\text{s.t. } 2k' - k \ge 3)$$

where $\mathbf{F}_{\text{EPZ}}(x_{u,v})$ and $\mathbf{F}_{\text{IFZ}}(x_{u,v})$ are the output of EPZ and IFZ convolution for the input representations $x_{u,v}$, $\mathbf{F}_{\text{IFZ}}(\cdot) = \mathbf{F}(\cdot)$. $s$ and $t$ denote the kernel position for the EPZ convolution. By

---

[2]For simplicity, we omit the channel-level operation, as it can be easily extended by following the standard convolutional settings.

leveraging the fine-grained perceptual discrepancies between EPZ and IFZ convolutions, $\mathbf{F}_c(x_{u,v})$ captures the intra-image distribution discrepancy at position $(u, v)$.

Then a category activation maps $\mathbf{C}$ is generated to eliminate ambiguous representations as follows:

$$\mathbf{C}_{u,v} = \begin{cases} 1, & |\mathbf{F}_c(x_{u,v})/\mathbf{F}_{\text{IFZ}}(x_{u,v})| < \lambda \\ 0, & |\mathbf{F}_c(x_{u,v})/\mathbf{F}_{\text{IFZ}}(x_{u,v})| \geq \lambda \end{cases} \tag{5}$$

where $\lambda$ is a hyperparameter for measuring ambiguous representations and $\mathbf{C} = \{\mathbf{C}_{u,v}|0 \leq u < h, 0 \leq v < w\}$. If the distribution discrepancy $\mathbf{F}_c(x_{u,v})$ for an representation $x_{u,v}$ is significantly smaller than the internal distribution $\mathbf{F}_{\text{IFZ}}(x_{u,v})$, this representation is activated, i.e., $\mathbf{C}_{u,v} = 1$. This indicates that $x_{u,v}$ belongs to the same category as its surrounding representations. Conversely, if a substantial discrepancy is observed, indicating an ambiguous boundary, $\mathbf{C}_{u,v}$ is set to 0.

Finally, local representations $\mathbf{F}_{\text{IFZ}}(\mathbf{X})$ of initial input representations $\mathbf{X}$ are multiplied by $\mathbf{C}$, that is $\mathbf{C} \cdot \mathbf{F}_{\text{IFZ}}(\mathbf{X})$, to eliminate ambiguous boundary representations $x^{ir}$, thereby facilitating a clear distinction of different regions within the image and further avoiding the oversegmentation of objects.

### 3.3 COMMONALITY-SPECIFICITY ATTENTION (CSA) MECHANISM

Subsequently, the CSA mechanism is proposed to learn the inter-image distribution discrepancies for mitigating the inference of similar structures by both the *Commonality Attention (CA)* and *Specificity Attention (SA)* mechanisms as in Figure 2(c).

**CA for Similarity Representations.** Representations $\mathbf{C} \cdot \mathbf{F}_{\text{IFZ}}(\mathbf{X})$ obtained from the C-Conv module go through CA in two distinct pathways, where one is for the pathological modality representations $\mathbf{C} \cdot \mathbf{F}_{\text{IFZ}}(\mathbf{X}_p)$ and the other is for the healthy modality representations $\mathbf{C} \cdot \mathbf{F}_{\text{IFZ}}(\mathbf{X}_h)$. $\mathbf{X}_p$ and $\mathbf{X}_h$ denotes the initial representations of the pathological and healthy image, respectively. For the first pathway, healthy modality representations $\mathbf{C} \cdot \mathbf{F}_{\text{IFZ}}(\mathbf{X}_h)$ are fed into an Average Buffer $ab(\cdot)$ (**Definition 3.1. in Appendix C**) to store a certain number of reference samples and compute their average distribution $ab(\mathbf{C} \cdot \mathbf{F}_{\text{IFZ}}(\mathbf{X}_h))$. This generates rich background structure information from the healthy modality. In this process, reference samples are dynamically replaced as the model updates. Subsequently, the SElayer algorithm[3] $s(\cdot)$Hu et al. (2018) is integrated to adaptively reweight the channel-wise average representations for increasing the expressive power of the healthy modality representations, forming $\mathbb{R}^{c \times hw}$ representation vectors $s(ab(\mathbf{C} \cdot \mathbf{F}_{\text{IFZ}}(\mathbf{X}_h)))$. Simultaneously, the second pathway takes the pathological modality representations $\mathbf{C} \cdot \mathbf{F}_{\text{IFZ}}(\mathbf{X}_p)$ and projects them to the consistent dimensional embedding space $\mathbb{R}^{c \times hw}$ via a reshape layer $reshape_1(\cdot)$, the reshaped pathological modality representations are denoted as $reshape_1(\mathbf{C} \cdot \mathbf{F}_{\text{IFZ}}(\mathbf{X}_p))$. Following this, an element-wise incidence matrix $\mathbf{R}$ is computed as:

$$\mathbf{R} = softmax(s(ab(\mathbf{C} \cdot \mathbf{F}_{\text{IFZ}}(\mathbf{X}_h)))^{\text{T}} \times reshape_1(\mathbf{C} \cdot \mathbf{F}_{\text{IFZ}}(\mathbf{X}_p))) \tag{6}$$

In $\mathbf{R}$, similarity regions between different modalities of images generally represent a common structural backbone with high probabilities falling in $[0.5, 1]$, i.e., commonality, whereas discrepancy regions between different modalities of images tend to approach zero, indicating the emergence of discriminative representations. Thus commonality attention maps $\mathbf{M}_c$ can be calculated by multiplying reshaped pathological modality representations with the incidence matrix $\mathbf{R}$:

$$\mathbf{M}_c = reshape_2(reshape_1(\mathbf{C} \cdot \mathbf{F}_{\text{IFZ}}(\mathbf{X}_p)) \times \mathbf{R}) \tag{7}$$

where $\mathbf{M}_c$ highlights the regions of similarity representations. $reshape_2(\cdot)$ resizes the commonality attention maps to the representation space $\mathbb{R}^{c \times h \times w}$ for further processing.

**SA for Discrepancy Representations.** Similarly, leveraging commonality attention maps $\mathbf{M}_c$, the specificity attention maps $\mathbf{M}_s$ can be calculated through multiplying pathological modality representations with an inverse activation projection $\mathbf{M}'_c$:

$$\mathbf{M}'_c(u, v) = \begin{cases} 1, & \text{softmax}(\mathbf{M}_c(u, v)) < 0.5 \\ 0, & \text{softmax}(\mathbf{M}_c(u, v)) \geq 0.5 \end{cases} \tag{8}$$

$$\mathbf{M}_s = \mathbf{C} \cdot \mathbf{F}_{\text{IFZ}}(\mathbf{X}_p) \cdot \mathbf{M}'_c \tag{9}$$

---

[3]More details about the SElayer algorithm are provided in Appendix D.

Specifically, when the similarity probability $\text{softmax}(\mathbf{M}_c(u,v))$ in commonality attention maps is below 50%, a potential discrepancy position is detected and the corresponding position of value in the inverse activation map $\mathbf{M}'_c(u,v)$ is set as 1. Conversely, $\text{softmax}(\mathbf{M}_c(u,v)) \geq 0.5$ indicates that a similar structure is found, $\mathbf{M}'_c(u,v) = 0$. Therefore, the resulting specificity attention maps identify local discriminative regions (discrepancy representations), which avoids the model's attention shifting away from target objects and hence mitigates the inference of similar structures. Furthermore, the commonality attention map $\mathbf{M}_c$ is added to the detected pathological modality representations $\mathbf{C} \cdot \mathbf{F}_{\text{IFZ}}(\mathbf{X}_p)$ to enhance the constraints of background structures and the specificity attention map $\mathbf{M}_s$ removes potential discrepancy distributions in pathological modality images to facilitate the representation distribution conversion across labels for further image-level supervision.

### 3.4 OBJECTIVE FUNCTION

Finally, CoinGAN devises a GAN-based adversarial loss function $\mathcal{L}_{adv}$ to complete this distribution conversion and further extract valuable regions of interest, thereby forming precise segmentation. Specifically, dual-enhanced representations $\mathbf{F}_{\text{CSA}}(\mathbf{C} \cdot \mathbf{F}_{\text{IFZ}}(\mathbf{X}_p))$ are fed into a *Generative Adversarial Network (GAN)*, comprising a generator $G(\cdot)$ and a discriminator $D(\cdot)$. The generator $G(\cdot)$ transforms these dual-enhanced representations towards a converted healthy image to fool the well-trained discriminator $D(\cdot)$. $D(\cdot)$ is trained to distinguish between different labels of medical images. The difference between the original image and the converted healthy image serves as the segmentation mask for the lesion region. This adversarial process is supervised by $\mathcal{L}_{adv}$:

$$\mathcal{L}_{adv} = -\sum_{u,v} \log(1 - D(G(\mathbf{F}_{\text{CSA}}(\mathbf{C} \cdot \mathbf{F}_{\text{IFZ}}(\mathbf{X}_p))))) + \log(D(\mathbf{I}_h)) \tag{10}$$

where $\mathbf{I}_h$ is the healthy image.

## 4 EXPERIMENTS

**Datasets & Metrics.** We conduct our extensive experiments on three public benchmarks: QaTa-COV19 Degerli et al. (2022) is a large-scale pneumonia benchmark dataset with 9,258 chest X-ray images. Both ISIC2018 Codella et al. (2019) and MoNuSeg Kumar et al. (2019) are the challenge datasets designed for skin lesions and kernel segmentation. ISIC2018 consists of 2,694 dermoscopy images while MoNuSeg involves 21,623 annotations on histopathologic images. More details about datasets are summarized in Appendix E. Six key metrics are utilized to assess the model performance, namely Dice Coefficient (DSC), Jaccard Coefficient (JC), Average Surface Distance (ASD), Accuracy (ACC), Specificity (SP), and Sensitivity (SE). Among them, the DSC, JC, and ASD metrics are specially utilized for measuring the precision of biomedical segmentation.

**Implementation Details.** We use CycleGAN Zhu et al. (2017) (a typical GAN) as the backbone without pre-trained weights, which consists of three stride-2 convolutions, nine residual blocks, and three $\frac{1}{2}$-strided convolutions. Our method is implemented in Python using the deep learning framework PyTorch and tested on a Tesla P40 GPU with 22GB of memory. In medical WSSS tasks, we use the stochastic gradient descent (SGD) optimizer with an initial learning rate of 2.5×1e-4 for the generator. The momentum is set to 0.9 and the weight decay is 5×1e-4. For the discriminator, we adopt the adaptive moment estimation (Adam) optimizer with an initial learning rate of 1e-4. Both learning rates are subjected to a polynomial decay scheduler with a decay power of 0.9. We train CoinGAN for 100 epochs for every dataset. In all experiments, we solely utilize image-level labels for supervision.

### 4.1 COMPARISON WITH SOTA BASELINES

For a comprehensive evaluation of CoinGAN for WSSS, we select eighteen state-of-the-art (SOTA) baselines for comparison, including AuxSegNet Xu et al. (2021), SESS Tursun et al. (2022), DRS Kim et al. (2021), Group Zhou et al. (2022), AffinityNet Ahn & Kwak (2018), IRNet Ahn et al. (2019), CONTA Zhang et al. (2020), RPNet Liu et al. (2021), SFC Zhao et al. (2024b), AMN Lee et al. (2022b), FPR Chen et al. (2023a), SeCo Yang et al. (2024b), MinMax Belharbi et al. (2021), WSSS-Tissue Han et al. (2022), OEEM Li et al. (2022), Swin-MIL Qian et al. (2022), CMER Patel & Dolz (2022), SA-MIL Li et al. (2023b).

Table 1: **CoinGAN surpasses SOTA WSSS methods on three application scenarios with 1.1~17.88%
improvement on comprehensive metrics.** The following table is the statistical comparison results on QaTa-
COV19. The last row "Fullsup" denotes results of a **full**y **sup**ervised segmentation method where we adopt a
standard segmentation model (U-Net) as a benchmark. I signifies the image-level supervision, S denotes the
use of saliency maps as additional auxiliary information. GT are the ground truth pixel-level annotations.

| | | | | | | | | |
|---|---|---|---|---|---|---|---|---|
| **QaTa-COV19** | | | | | | | | |
| **Method** | **Backbone** | **Sup.** | **DSC(%)↑** | **JC(%)↑** | **ASD↓** | **ACC(%)↑** | **SP(%)↑** | **SE(%)↑** |
| **Image-level supervision + Saliency map.** | | | | | | | | |
| AuxSegNet | ResNet38 | I+S | 60.07 | 44.90 | 2.99 | 66.73 | 62.46 | 93.21 |
| DRS | VGG16/ResNet101 | I+S | 60.56 | 46.89 | 2.61 | 72.18 | 73.43 | 64.43 |
| Group | VGG16/ResNet101 | I+S | 63.44 | 49.01 | 2.17 | 72.18 | 70.30 | 83.83 |
| SESS | VGG16/ResNet50 | I+S | 65.36 | 52.00 | 1.72 | 77.25 | 78.87 | 67.20 |
| **Image-level supervision.** | | | | | | | | |
| AffinityNet | ResNet38 | I | 33.16 | 19.88 | 7.11 | 33.17 | 20.49 | **99.72** |
| IRNet | ResNet50 | I | 52.55 | 38.54 | 3.03 | 62.14 | 62.18 | 61.87 |
| CONTA | ResNet50/ResNet101 | I | 53.37 | 40.01 | 3.25 | 65.25 | 67.19 | 53.21 |
| RPNet | ResNet50/ResNet101 | I | 56.19 | 42.03 | 2.76 | 66.03 | 65.85 | 67.11 |
| AMN | ResNet50 | I | 55.69 | 42.19 | 3.06 | 67.58 | 69.32 | 56.82 |
| FPR | ResNet50 | I | 64.82 | 51.53 | 2.42 | 77.08 | 79.02 | 65.01 |
| SeCo | ViT-B/16 | I | 61.83 | 50.52 | 3.44 | 81.01 | **95.18** | 25.05 |
| SFC | ResNet101 | I | 53.88 | 39.57 | 3.07 | 62.81 | 90.43 | 22.97 |
| **CoinGAN (ours)** | CycleGAN | I | **71.69** | **58.71** | **1.45** | **82.11** | 82.90 | 77.20 |
| Fullsup | U-Net | GT | 80.76 | 70.11 | 0.82 | 91.13 | 95.53 | 63.78 |

Table 2: Statistical comparison with SOTA WSSS methods on ISIC2018.

| | | | | | | | | |
|---|---|---|---|---|---|---|---|---|
| **ISIC2018** | | | | | | | | |
| **Method** | **Backbone** | **Sup.** | **DSC(%)↑** | **JC(%)↑** | **ASD↓** | **ACC(%)↑** | **SP(%)↑** | **SE(%)↑** |
| **Image-level supervision + Saliency map.** | | | | | | | | |
| AuxSegNet | ResNet38 | I+S | 82.74 | 71.44 | 1.22 | 86.81 | 92.61 | 71.06 |
| DRS | VGG16/ResNet101 | I+S | 76.95 | 63.07 | 1.59 | 79.10 | 74.98 | 90.27 |
| Group | VGG16/ResNet101 | I+S | 82.08 | 70.64 | 1.27 | 86.79 | 94.48 | 65.93 |
| SESS | VGG16/ResNet50 | I+S | 80.91 | 69.09 | 1.23 | 85.97 | 94.02 | 64.10 |
| **Image-level supervision.** | | | | | | | | |
| AffinityNet | ResNet38 | I | 63.47 | 47.42 | 3.95 | 66.70 | 59.35 | **98.52** |
| IRNet | ResNet50 | I | 76.47 | 63.02 | 1.59 | 80.93 | 85.15 | 69.45 |
| CONTA | ResNet50/ResNet101 | I | 77.78 | 65.22 | 1.56 | 84.21 | 94.42 | 56.50 |
| RPNet | ResNet50/ResNet101 | I | 82.23 | 70.83 | 1.23 | 86.84 | 94.23 | 66.75 |
| AMN | ResNet50 | I | 80.61 | 68.57 | 1.27 | 85.18 | 91.47 | 68.10 |
| FPR | ResNet50 | I | 82.45 | 70.72 | 1.31 | 85.13 | 85.05 | 85.39 |
| SeCo | ViT-B/16 | I | 82.55 | 71.00 | 1.3 | 85.88 | 91.75 | 71.79 |
| SFC | ResNet101 | I | 80.51 | 68.85 | 1.37 | 86.89 | 85.64 | 93.57 |
| **CoinGAN (ours)** | CycleGAN | I | **84.73** | **74.27** | **1.20** | **88.57** | **94.92** | 71.33 |
| Fullsup | U-Net | GT | 88.37 | 79.63 | 1.13 | 91.13 | 95.66 | 78.83 |

Table 3: Statistical comparison with SOTA WSSS methods on MoNuSeg.

| | | | | | | | | |
|---|---|---|---|---|---|---|---|---|
| **MoNuSeg** | | | | | | | | |
| **Method** | **Backbone** | **Sup.** | **DSC(%)↑** | **JC(%)↑** | **ASD↓** | **ACC(%)↑** | **SP(%)↑** | **SE(%)↑** |
| **Image-level supervision + Saliency map.** | | | | | | | | |
| AuxSegNet | ResNet38 | I+S | 38.30 | 24.95 | 1.19 | 42.60 | 12.43 | 98.86 |
| DRS | VGG16/ResNet101 | I+S | 50.88 | 36.13 | 1.05 | 57.58 | 72.61 | 29.56 |
| Group | VGG16/ResNet101 | I+S | 42.70 | 27.31 | 3.94 | 43.28 | 25.51 | 76.40 |
| SESS | VGG16/ResNet50 | I+S | 60.89 | 43.94 | 0.33 | 61.45 | 56.34 | 70.98 |
| **Image-level supervision.** | | | | | | | | |
| AffinityNet | ResNet38 | I | 19.27 | 11.93 | 8.97 | 23.87 | 2.59e-05 | **99.99** |
| IRNet | ResNet50 | I | 49.60 | 33.29 | 1.37 | 50.65 | 49.99 | 51.87 |
| CONTA | ResNet50/ResNet101 | I | 50.07 | 33.85 | 3.15 | 51.61 | 53.13 | 48.77 |
| RPNet | ResNet50/ResNet101 | I | 52.62 | 37.11 | 1.04 | 57.33 | 58.35 | 54.06 |
| AMN | ResNet50 | I | 53.31 | 38.74 | 1.01 | 61.30 | 67.42 | 41.76 |
| FPR | ResNet50 | I | 52.21 | 38.02 | 0.91 | 61.14 | 68.94 | 36.83 |
| SeCo | ViT-B/16 | I | 61.21 | 46.78 | 0.84 | 70.26 | 71.24 | 65.38 |
| SFC | ResNet101 | I | 48.61 | 33.00 | 0.76 | 51.60 | 64.14 | 33.53 |
| **CoinGAN (ours)** | CycleGAN | I | **78.32** | **64.66** | **0.31** | **79.54** | **80.02** | 79.28 |
| Fullsup | U-Net | GT | 81.95 | 69.85 | 0.33 | 83.89 | 89.59 | 73.25 |

Table 4: Statistical comparison with recent medical WSSS methods on QaTa-COV19.

Table 5: Statistical comparison with recent medical WSSS methods on MoNuSeg.

| QaTa-COV19 | | | | MoNuSeg | | | |
|---|---|---|---|---|---|---|---|
| **Method** | **DSC(%)** | **JC(%)** | **ACC(%)** | **Method** | **DSC(%)** | **JC(%)** | **ACC(%)** |
| MinMax | 46.60 | 32.11 | 52.38 | MinMax | 47.17 | 31.62 | 49.70 |
| WSSS-Tissue | 31.79 | 19.19 | 32.81 | WSSS-Tissue | 40.31 | 32.49 | 63.82 |
| OEEM | - | 25.38 | 60.01 | OEEM | - | 30.01 | 61.25 |
| Swin-MIL | 42.03 | 27.22 | 44.09 | Swin-MIL | 34.66 | 23.31 | 42.63 |
| CMER | 63.03 | 50.86 | 79.32 | CMER | 60.99 | 44.02 | 61.48 |
| SA-MIL | 52.03 | 41.70 | 73.10 | SA-MIL | 50.80 | 34.78 | 53.25 |
| **CoinGAN (ours)** | **71.69** | **58.71** | **82.11** | **CoinGAN (ours)** | **78.32** | **64.66** | **79.54** |

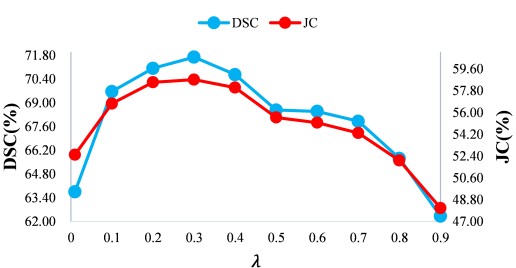

**Performance Drop**

**Parameter sensitivity analysis**

Figure 3: **There is a serious performance drop of SOTA baselines between natural images (orange) and medical images (blue).** This substantiates the more challenging characteristics of medical images in the field of pathological segmentation.

Figure 4: $\lambda = 0.3$ **achieves the best performance.** The hyperparameter varying from 0.01 to 0.9 is analysed in CoinGAN based on QaTa-COV19 using the DSC(%) (Left ordinate axis) and JC(%) (Right ordinate axis) metrics.

**Comparison with General SOTA Baselines.** Comparison results with the general SOTA WSSS methods are present in Tables 1, 2, 3. CoinGAN achieves the best performance across three datasets. On the ISIC2018 dataset, CoinGAN achieves an improvement with a 2.83% JC and 1.99% DSC gain against the SOTA method. Further, larger performance improvements are achieved on the QaTa-COV19 and MoNuSeg datasets, including the pneumonia segmentation with a 6.71% JC and 6.33% DSC gain (QaTa-COV19) and kernel segmentation with a 17.88% JC and 17.11% DSC gain (MoNuSeg). The main reason for the performance improvement gap is extracorporeal lesions, e.g., skin cancer, are exposed to air and usually have a clear boundary, which makes them easy to distinguish. However, glass-frosted pneumonia lesions and kernels have a close relationship with the chest structure and tissues because they depend on nutrients provided by these structures, resulting in the complexity of segmentation tasks, e.g., vague boundaries, and the connectivity of foreground and background. Thus all these prove the superiority of our method in resolving ambiguous boundaries and structural constraints.

**Comparison with Domain-specific SOTA Baselines.** Apart from the general SOTA WSSS methods, we also assess CoinGAN against medical SOTA methods, particularly for more challenging internal medical scenarios, namely QaTa-COV19 and MoNuSeg. As depicted in Tables 4 and 5, CoinGAN exhibits the best performance across evaluation metrics and application scenarios. On the contrary, existing medical WSSS methods are not satisfactory. This is because they are primarily designed for some specific organs or scenarios, and at times necessitate additional auxiliary information, e.g., patch-level annotation, which significantly limits their application scenarios as discussed in Sec. 2. In contrast, CoinGAN extracts the discrepancy information directly from medical images themselves, enhancing its effectiveness across various application scenarios.

## 4.2 PERFORMANCE DROP ON DIFFERENT DATA TYPES

As in Figure 3, we also compare the performance of recent SOTA WSSS methods in medical images and natural images. It can be seen that there is a serious performance drop from natural images (PASCAL VOC2012) to medical images (QaTa-COV19). This performance decline suggests that medical images indeed possess some unique and more challenging characteristics that make these methods fall into a dilemma when segmenting pathological regions, e.g., ambiguous boundaries, distinctive distribution patterns and similar structures.

(a) Skin Lesion Segmentation       (b) Pneumonia segmentation

Figure 5: **Visualized results validate CoinGAN achieves more boundary-aware and accurate segmentation.** Results of CoinGAN and best-performing WSSS methods on ISIC2018 and QaTa-COV19 are presented.

### 4.3 ABLATION STUDY AND VISUALIZATION STUDY

To investigate the effectiveness of each component, we perform an ablation study on three major modules: the C-Conv module, the CA and SA mechanism. Besides, we also explore the ability of saliency maps (**S**) to constrain output results as in Tursun et al. (2022). All experiments are conducted on the QaTa-COV19 dataset.

Ablation study results are shown in Table 6, with each row corresponding to a different experimental setup. Specifically, the first row denotes the backbone result as a reference. The second row introduces the C-Conv module alone. This leads to a significant improvement (3.14% DSC gain) over the backbone, indicating that the C-Conv module effectively enhances intra-image

Table 6: **Ablation studies indicate that the C-Conv module and CSA mechanism achieved the most significant improvements.** Besides, each component is individually investigated in CoinGAN based on QaTa-COV19.

| QaTa-COV19 | | | | | |
|---|---|---|---|---|---|
| Backbone | C-Conv | CA | SA | S | DSC(%) |
| ✓ | | | | | 65.15 |
| ✓ | ✓ | | | | 68.29 |
| ✓ | ✓ | ✓ | | | 70.82 |
| ✓ | ✓ | | ✓ | | 71.10 |
| ✓ | ✓ | ✓ | ✓ | | 71.69 |
| ✓ | ✓ | ✓ | ✓ | ✓ | 72.67 |

distribution and contributes to cleaner representations. The third to fifth rows validate the performance of the CA, SA, and CSA mechanisms. The CA mechanism strengthens the common background structure. This aids in cleaning up false positive samples, facilitating a notable improvement. The SA mechanism removes the discrepancy distribution which directly rectifies the mistaken negative (true positive) samples for further improvement. Notably, the SA mechanism outperforms CA. This is because semantic segmentation focuses more on accurate targets (true positive samples) compared with common structural background constraints. Finally, the integrated CSA mechanism achieves the best overall performance. Besides, salient maps can further improve our model performance with a 0.98% increase in DSC. Visualization results (Figure 5) validate our CoinGAN achieves more boundary-aware and accurate segmentation results (More details in Appendix H).

### 4.4 HYPERPARAMETER SENSITIVITY STUDY

To analyze the influence of the hyperparameter $\lambda$ in the C-Conv module, we conduct experiments with different $\lambda$ values while keeping other parameters fixed. As depicted in Figure 4, overly large $\lambda$ values relax the limitations on boundary regions, introducing more noise to representation learning and hence limiting the model performance. Conversely, overly small $\lambda$ values render the convolution oversensitive to changes within images. Even minor changes are mistaken for the appearance of a boundary, causing the category activation maps $M_{c\_conv}$ to filter out too many mistaken ambiguous representations, which results in a loss of information. $\lambda = 0.3$ achieves the best performance, striking a balance between sensitivity to boundaries and retaining medical information.

### 5 CONCLUSIONS

In this paper, we propose CoinGAN, a novel model to address challenges posed by medical WSSS. CoinGAN features two pivotal modules: the **C-Conv Module** and the **CSA Mechanism**, to unearth latent knowledge inherent in the images themselves. The C-Conv module leverages intra-image representation perceptual discrepancies to eliminate the boundary uncertainty, while the CSA mechanism exploits inter-image representation discrepancies for precise pathological segmentation. Comprehensive experiments substantiate the efficacy of CoinGAN across diverse datasets.

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

# APPENDIX

## A CLARIFICATIONS OF CONCEPTS

- Oversegmentation refers to instances where the segmentation results extend beyond the actual ground truth object region, incorrectly including background areas as part of the object (false positives). This results in a segmentation "overflow."
- Inaccurate shapes describe cases where the segmented regions deviate significantly from the true shapes of the target objects, failing to align with the ground truth lesions and leading to segmentation errors that misrepresent the actual lesion boundaries.

## B DETAILED ILLUSTRATIONS FOR FIGURE 1

Figure 1 (first row) illustrates the probability mass function (PMF) for two different labels of medical images. The horizontal axis represents the intensity values of image pixels (ranging from 0 to 255), while the vertical axis denotes the probability of each intensity value. Specifically:

- Intensity refers to the numerical value of a pixel's intensity.
- Distribution represents the PMF of intensity values in medical images.

## C AVERAGE BUFFER

**Definition 3.1.** *Average Buffer.* Average Buffer $ab(\cdot)$ is a sample buffer designed to store a certain number of reference samples $\mathcal{A} = \{\mathbf{A}_1, \mathbf{A}_2, \ldots, \mathbf{A}_N\}$ and compute their average representations $\bar{\mathbf{A}}$. During the model training, these stored samples are dynamically updated with each batch of input samples $\{\mathbf{A}'_1, \mathbf{A}'_2, \ldots, \mathbf{A}'_N\}$. $N$ is the batch size.

To clarify this process, the detailed mathematical definition is formulated as follows:

Let:

- $\mathcal{A} = \{\mathbf{A}_1, \mathbf{A}_2, \ldots, \mathbf{A}_N\}$ represent the set of $N$ stored reference samples in the Average Buffer, where $N$ is the batch size.
- $\mathbf{A}_i$ denote the $i$-th sample in the Average Buffer.

The Average Buffer computes the average representation $\bar{\mathbf{A}}$ of the stored samples as:

$$\bar{\mathbf{A}} = \frac{1}{N} \sum_{i=1}^{N} \mathbf{A}_i \tag{11}$$

where $\mathbf{A}_i$ refers to the representation derived from the C-Conv module, specifically $\mathbf{C} \cdot \mathbf{F}_{\text{IFZ}}(\mathbf{X}_h)$.

During model training, when a new batch of samples $\{\mathbf{A}'_1, \mathbf{A}'_2, \ldots, \mathbf{A}'_N\}$ is input to the Average Buffer, the existing samples $\{\mathbf{A}_1, \mathbf{A}_2, \ldots, \mathbf{A}_N\}$ are dynamically replaced by the new batch $\{\mathbf{A}'_1, \mathbf{A}'_2, \ldots, \mathbf{A}'_N\}$.

After replacement, the updated Average Buffer is:

$$\mathcal{A}_{new} = \{\mathbf{A}'_1, \mathbf{A}'_2, \ldots, \mathbf{A}'_N\} \tag{12}$$

and the average representation of the new Average Buffer becomes:

$$\bar{\mathbf{A}}_{new} = \frac{1}{N} \sum_{i=1}^{N} \mathbf{A}'_i \tag{13}$$

where $\mathbf{A}'_i$ is the representation computed by the C-Conv module for the new batch of samples.

In the CSA mechanism, the Average Buffer enables CoinGAN to capture rich structural background information from the healthy modality, using the computed average representation as a reference.

# D  SELAYER

The SElayer is a channel-wise adaptive weighting algorithm originally proposed in Hu et al. (2018). The SElayer enables the neural network to prioritize the most critical features for the task at hand, boosting the expressive capacity of representations. In CoinGAN, we integrate the SElayer following the Average Buffer component to adaptively reweight the channel-wise average representations $\bar{\mathbf{A}}$, thereby increasing the expressive power of the healthy modality reference representation and improving the utilization of background information in subsequent processing. To clarify, the technical details of the SElayer algorithm are outlined as follows:

---

**Algorithm 1** SELayer (Squeeze-and-Excitation Layer)

---

**Require:** Channel-wise average representations $\bar{\mathbf{A}}$ with shape $(N_H, N_W, N_L)$, where $N_H$ is the height, $N_W$ is the width, and $N_L$ is the number of channels.
1: Compute the channel descriptor $\mathbf{z}$ via global average pooling with Eq. 14
2: Pass $\mathbf{Z}$ through a bottleneck architecture comprising two fully connected layers with Eq. 15
3: Recalibrate the input representations $\bar{\mathbf{A}}$ by element-wise scaling with $\hat{\mathbf{Z}}$ with Eq. 16
4: Return the recalibrated representations $\mathbf{B}$

---

1. Compute the channel descriptor $\mathbf{Z}$ of the input representations $\bar{\mathbf{A}}$ via global average pooling:

$$\mathbf{Z}_l = \frac{1}{N_H \times N_W} \sum_{i=1}^{N_H} \sum_{j=1}^{N_W} \bar{\mathbf{A}}_{ijl}, \forall l = 1, 2, \ldots, N_L \qquad (14)$$

where $N_H$, $N_W$, and $N_L$ are the height, width, and number of channels, respectively, of the representations $\bar{\mathbf{A}}$.

2. Pass the channel descriptor $\mathbf{Z}$ through a bottleneck architecture comprising two fully connected layers for learning the channel-wise weights:

$$\hat{\mathbf{Z}} = \sigma(\mathbf{W_2}(\text{ReLU}(\mathbf{W_1}\mathbf{Z} + \mathbf{b_1})) + \mathbf{b_2}) \qquad (15)$$

where $\sigma$ denotes the sigmoid activation function, and $\hat{\mathbf{Z}}$ is the recalibration vector with shape $(1, 1, N_L)$.

3. Recalibrate the input representations $\bar{\mathbf{A}}$ by element-wise scaling with $\hat{\mathbf{Z}}$:

$$\mathbf{B}_{ijl} = \bar{\mathbf{A}}_{ijl} \cdot \hat{\mathbf{Z}}_l, \forall i, j, l \qquad (16)$$

4. Return the recalibrated representations $\mathbf{B}$.

This integration of the SElayer refines the CSA mechanism by emphasizing the most relevant features, facilitating the effective exploitation of background information. The SElayer algorithm is summarized in Algorithm 1.

# E  DATASETS

**Datasets.** To evaluate the effectiveness of CoinGAN, we conduct experiments for medical WSSS tasks on three public benchmarks, including QaTa-COV19 Degerli et al. (2022), ISIC2018 Codella et al. (2019), and MoNuSeg Kumar et al. (2019).

**QaTa-COV19** Degerli et al. (2022) is a large-scale benchmark **COV**ID-**19** dataset compiled by **Qa**tar University and **Ta**mpere University. It contains 9,258 COVID-19 chest X-rays with ground truth segmentation masks (pneumonia lesions) and 12,544 healthy chest X-rays as the control group that have a resolution of $224 \times 224$ pixels. To ensure a fair comparison with state-of-the-art methods, we adopt the same experimental setting as Yamac et al. (2021): 80% samples are used for training, and 20% samples for testing.

**ISIC2018** Codella et al. (2019) is a large-scale skin lesion segmentation challenge dataset, comprising dermatoscopic data collected from multiple treatment centers. Images vary in size ranging from

556 × 679 to 4499 × 6748 pixels with the corresponding skin lesion annotations. For the healthy images, inspired by Tschandl et al. (2020), we crop healthy skin regions from the backgrounds of these images and apply bilinear interpolation, resulting in healthy images (control samples). Following the previous work Zhang et al. (2022), we resize all the images and their masks by the bilinear interpolation and split this dataset into a training set with 2,594 dermoscopy images and a test set with 100 images. (Additionally, there are 1,000 extra samples that are unavailable on the ISIC challenge website: https://challenge.isic-archive.com/data/#2018.)

**MoNuSeg** Kumar et al. (2019) is a MICCAI 2018 Challenge dataset designed for the multi-instance segmentation task. It consists of 21,623 single kernel annotations in the histopathologic images of H&E (Hematoxylin and Eosin) stained tissue, with a resolution of 1000 × 1000 pixels for each image. Similar to the approach in Yang et al. (2024a), we process this dataset to derive healthy images by applying morphological operations (image erosion) and Gaussian filtering to obtain the background information. The segmented nuclear status can be used for assessing cancer grade and treatment effectiveness. Following a strategy similar to the previous work Wu et al. (2022), we crop images sequentially for data augmentation. Among them, 80% samples are used as training images and 20% samples are used as evaluation images.

**BraTS 2021** (Brain Tumor Segmentation 2021) Bakas et al. (2017; 2018); Menze et al. (2014) is a widely used benchmark dataset for the segmentation of brain tumors from MRI scans. It contains 2,000 3D brain scans, each of which includes four different MRI modalities (e.g., T1, T1ce, T2, and FLAIR) as well as tumor segmentation ground truth. The official data divides these cases by the ratio of 8:1:1 for training, validation, and testing. Following the latest practices Chen et al. (2023b); Hu et al. (2023); Hatamizadeh et al. (2021), the FLAIR channel is used for the model training and validation, then the validation set is used to evaluate the model performance for a fair comparison.

**MELA** Wang et al. (2022) is a large-scale benchmark dataset for the mediastinal lesion analysis. It contains a training set with 770 Computed Tomography (CT) scans, a validation set with 110 CT scans, and a test set with 220 CT scans. Each CT slice has a resolution of 512 × 512 pixels. A total of 1,152 mediastinal lesions are annotated using the bounding boxes (diameter: 10–204 mm, mean: 48 mm). While MELA does not provide precise segmentation masks, it provides valuable bounding boxes with the size information of lesions which indicates the disease severity, rendering it suitable for the prospective research - the disease evolution.

## F  IN-DEPTH ANALYSES OF COMPARATIVE EXPERIMENTS

The comparative experimental results are presented in Tables 1 and 2 in the main paper. We analyze the model performance from two key perspectives: metric analysis and scenario analysis.

**Metric Analysis.** 1) Our CoinGAN achieves the best performance in all comprehensive evaluation metrics on three datasets. This validates the effectiveness of CoinGAN in leveraging the distinctive medical distribution divergence to mine the latent knowledge gap among different pathological modalities. 2) We note that AffinityNet Ahn & Kwak (2018) achieves the highest SE but yields inferior results in other metrics across all three datasets. This is because AffinityNet propagates local responses to nearby areas based on semantic affinities, leading to representations that are biased towards the positive category of positions and some false positive positions with high biological affinities. This results in a high SE but low SP in the medical application scenarios. AuxSegNet Xu et al. (2021) presents similar issues but its performance is slightly rectified by extra auxiliary information, e.g., saliency maps (S). On the other hand, some methods overly rely on image-level classification models, and the unique distribution patterns of medical images lead to rapid convergence through shortcuts, further resulting in incomplete exploration of discriminative regions (high SP, low SE), such as Seco. 3) Our CoinGAN demonstrates the closest performance to the ***Fully supervised semantic segmentation (Fullsup)***, even keeps similar metric distributions. This suggests that the knowledge discrepancy underlying medical modalities holds promising potential to compete with manual annotations. Medical images inherently contain a wealth of information. 4) We note that certain models rely on extra auxiliary information (S+I) to improve their performance, yet are not effective across all application scenarios, e.g., MoNuSeg. Our PAIR effectively eliminates the need for auxiliary information (S) and achieves superior performance solely using image-level annotations (I).

**Scenario Analysis.** We discuss this part from two standpoints: the extracorporeal scenario and the internal scenario. In the extracorporeal WSSS scenario, CoinGAN achieves a 2.83% boost in JC and a 1.99% boost in DSC over the best-performing WSSS method on the ISIC2018 dataset. In this context, extracorporeal lesions, e.g., skin cancer, are exposed to air and typically have well-defined boundaries like natural scenarios, enabling them to work well in lesion segmentation, but our CoinGAN still achieves a slight improvement. Simultaneously, in the internal WSSS scenarios, CoinGAN demonstrates more significant performance improvements. Specifically, in pneumonia segmentation on the QaTa-COV19 dataset, CoinGAN substantially achieves a 6.71% rise in JC and a 6.33% rise in DSC. Additionally, in kernel segmentation on the MoNuSeg dataset, CoinGAN delivers remarkable improvements with a 17.88% increase in JC and a 17.11% increase in DSC. Such performance improvement can be attributed to the unique characteristics of medical images. Glass-frosted pneumonia lesions, for example, depend on nutrients provided by chest cavity tissues, having a tight relationship with the chest structure, which results in ambiguous boundaries. Symmetrical structures, besides, give rise to structural interference. Similarly, in histopathologic tissues, tissues, and kernels are often intertwined, with different developmental cycles contributing to more complex dependencies and elusive structural relationships. The capability of CoinGAN to address these challenges makes it more effective in intra-body medical WSSS scenarios.

## G  VISUALIZATION STUDY FOR DISTRIBUTION CONVERSION

Additionally, we also demonstrate the efficacy of the distribution conversion through comparative distribution maps as depicted in Figure 6. The initial distribution maps (Red) reveal a pronounced concentration of sharp lesion signals. After implementing the distribution conversion, the distribution maps (Blue) display an adaptive redistribution from the sharp probability distribution pattern to a homogeneous probability distribution pattern. Such distribution conversion confirms that CoinGAN successfully grasps the latent knowledge gap, that is, the separated lesion area.

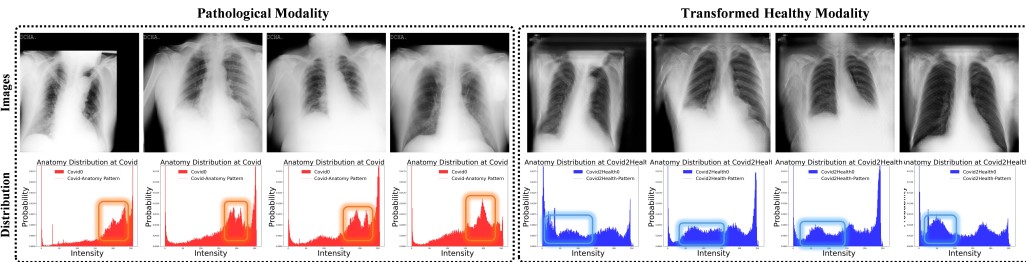

Figure 6: **The distribution conversion confirms that our method has indeed grasped the knowledge gap underlying distribution discrepancies, further ensuring lesion segmentation.** Some visualized distribution maps are presented for comparison. Left: The original pathological X-rays and the corresponding distribution maps (Red); Right: The converted images and the corresponding distribution maps after the distribution conversion (Blue).

## H  VISUALIZATION STUDY

Figure 5 provides an intuitive visualization demonstration. With the introduction of the C-Conv module and CSA mechanism, CoinGAN provides clearer boundaries and more accurate segmentation shapes. This demonstrates that CoinGAN well addresses the segmentation overflow noise and the latent incomplete exploration of lesion regions. More true positive lesions and fewer false positive backgrounds make CoinGAN reach the state-of-the-art WSSS performance, generating dense and high-quality pseudo annotations.

## I  EVALUATION OF COINGAN ON A DIFFERENT MEDICAL IMAGING MODALITY AND LESS COMMON DISEASES

To investigate the performance of CoinGAN on other forms of medical imaging data and less common diseases, we include the BraTS 2021 Bakas et al. (2017; 2018); Menze et al. (2014) dataset in our experiments for the detection and segmentation of brain tumors. This dataset focuses on MRI

images and includes diverse cases of brain tumors, such as gliomas and other less common tumor types. We evaluate CoinGAN on this dataset to demonstrate its generalizability to the MRI modality and its applicability to less common disease types such as gliomas. The experimental results are presented in Figure 7:

Table 7: **Evaluation of CoinGAN on the BraTS 2021 dataset.**

| Method | DSC(%)↑ | JC(%)↑ | ASD↓ | ACC(%)↑ | SP(%)↑ | SE(%)↑ |
|---|---|---|---|---|---|---|
| | | | **BraTS 2021** | | | |
| SeCo | 69.68 | 61.76 | 2.04 | 98.38 | 99.24 | 38.39 |
| FPR | 86.07 | 77.80 | 0.67 | 97.88 | 98.80 | **74.91** |
| **CoinGAN** | **89.41** | **82.30** | **0.47** | **98.56** | **99.63** | 72.14 |

The experimental results show that CoinGAN achieves superior performance, surpassing state-of-the-art image-level weakly supervised semantic segmentation models, including FPR and SeCo. Notably, FPR and SeCo were previously the top-performing models across the three datasets, as detailed in Tables 1, 2 and 3.

## J COMPARISON WITH A STATE-OF-THE-ART DOMAIN-SPECIFIC DIFFUSION-BASED MODEL

As a new baseline for comparison. CG-CDM Hu et al. (2023) is specifically tailored for medical WSSS, making it a suitable counterpart for our study. In the CG-CDM paper, the BraTS dataset is used to assess model performance, with image-level labels for training, and the reported results of CG-CDM are retrieved from [1]. Additionally, we have evaluated our proposed CoinGAN model on the same BraTS dataset. The results are summarized in Table 8:

Table 8: **The comparison with a state-of-the-art domain-specific diffusion-based model on the BraTS 2021 dataset.**

| Method | DSC(%)↑ | JC(%)↑ |
|---|---|---|
| | **BraTS 2021** | |
| CG-CDM | 56.3 | 45.0 |
| **CoinGAN** | **89.41** | **82.30** |

The experimental results highlight the superior performance of CoinGAN, further validating the effectiveness of our proposal.

## K COINGAN FOR MODELING THE PROGRESSION OF MEDIASTINAL LESIONS

To explore the severity levels as image-level labels in medical WSSS, we introduce a new dataset, MELA Wang et al. (2022) (more details elaborated in Appendix E), designed to analyze the progression of mediastinal lesions. In this dataset, medical images are categorized based on lesion sizes, with smaller lesions labeled as mild and larger ones as severe. This labeling reflects object sizes rather than presence/absence. Using this dataset, we conduct extensive visualization studies to investigate the conversion from mild to severe lesions, effectively simulating the progression of mediastinal lesions. The visualized results are presented in Figure 7 with the white box highlighting the mediastinal region.

**Transition from Mild to Severe Disease State:** This task represents the worsening of a medical condition, where lesions gradually become more severe. From the process of lesion expansion, it can be observed that the mediastinum gradually enlarges, covering normal tissues and expanding into unobstructed spaces. This expansion of the mediastinum compresses surrounding tissues and can compromise the respiratory system in the human body. As diseases progress, CoinGAN allows for the assessment of potential harm, which is crucial for developing the treatment plan. Further, this

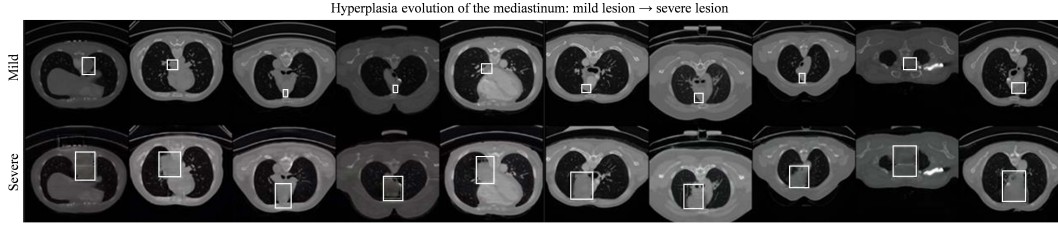

Figure 7: **The visualized results present the progression of mediastinal lesions from mild to severe lesions.** The white box highlights the mediastinal region.

exploratory study casts light on the potential of CoinGAN in facilitating smooth transitions between varying-degree modalities, which is invaluable in medical imaging and diagnosis.

## L   ERROR ANALYSIS OF COINGAN

We perform the error analysis by elaborating on the core idea of CoinGAN - the utilization of discrepancy information. The discrepancies between different labels of medical images are pivotal for distinguishing discriminative regions (i.e., regions of interest), forming the foundation of our proposed CoinGAN. CoinGAN leverages this discrepancy information to enhance the segmentation of target objects in medical images.

However, discrepancy information can be affected by potentially "inaccurate" image-level labels, which are not uncommon in clinical practice. For instance, some individuals labeled as healthy controls may exhibit subtle abnormalities in their medical images that resemble patient lesions but have not progressed to a diagnosable disease stage. These "healthy" images may introduce ambiguity into the model's learning process, impairing the effective use of discrepancy information and, consequently, the overall model performance.

Moreover, such data quality issues stemming from the complexity of real-world clinical scenarios can also affect the performance of other weakly supervised semantic segmentation methods, as evidenced by the robustness study in Appendix M. Addressing these challenges remains a critical area for future exploration.

## M   ROBUSTNESS STUDY OF COINGAN TO INACCURACIES AND VARIABILITY IN IMAGE-LEVEL LABELS

To assess the robustness of CoinGAN to inaccuracies and variability in image-level labels, we design and supplement the following experiment. Specifically, following the protocol of existing robustness studies such as Wei et al. (2021), we intentionally introduce inaccuracies in 10% of the image-level labels and evaluate the performance of CoinGAN alongside the best-performing weakly supervised semantic segmentation baseline method under these conditions.

For this robustness study, we use the QaTa-COV19 dataset as the reference. The experimental results, including a comparison of CoinGAN with the baseline, are summarized in Table 9.

Table 9: **The robustness study to inaccuracies and variability in image-level labels on the QaTa-COV19 dataset**

| | QaTa-COV19 | | | |
|---|---|---|---|---|
| **Method** | **DSC(%)↑** | **JC(%)↑** | **ASD↓** | **ACC(%)↑** |
| FPR-10% | 63.20 (-1.62) | 48.99 (-2.54) | 2.70 (-0.28) | 72.74 (-4.34) |
| CoinGAN-10% | 70.04 (-1.65) | 57.19 (-1.52) | 2.03 (-0.58) | 81.88 (-0.23) |

The values on the left represent the models' performance with 10% inaccurate image-level labels, while the values in parentheses indicate performance changes relative to the scenario without label

inaccuracies. "-" signifies a decline in performance. Detailed results for the scenario without label inaccuracies are provided in Table 1 of the manuscript. Notably, FPR is identified as the best-performing weakly supervised semantic segmentation model on the QaTa-COV19 dataset under image-level labels, as shown in Table 1.

The experimental results reveal that both CoinGAN and the baseline method experience performance degradation across all four comprehensive metrics when subjected to inaccurate labels. However, CoinGAN exhibits smaller or comparable performance variations in three key metrics—DSC, JC, and ACC—compared to the baseline, highlighting its superior robustness against label inaccuracies.

## N    DISCUSSION ON RELATED LITERATURE

**Discussion about the first challenge (distribution discrepancies):** Most of the generative method-based approaches can effectively address inherent distribution discrepancies. Specifically, we propose to use the generative adversarial network (GAN) within the CoinGAN model to exploit such discrepancies. Notably, some diffusion models, e.g., Hu et al. (2023); Li et al. (2023a); Liang et al. (2023); Gonzalez-Jimenez et al. (2023) can present viable alternatives as backbones, which represents a promising avenue for further exploration.

**Discussion about the second challenge (ambiguous boundaries):** We discuss three related studies, including BoundaryCAM Prabakaran et al. (2023), CTO Lin et al. (2023a), and boundary-aware CNNs Hatamizadeh et al. (2019). Below, we summarize the working principles of these studies with boundary-aware modules:

- BoundaryCAM Prabakaran et al. (2023) employs an unsupervised clustering strategy to extract clusters of pixels, which assist in defining an initial boundary of the target object. Subsequently, BoundaryCAM combines Class Activation Mapping (CAM) with the Floodfill Fishkin & Barsky (1985) algorithm to refine this initial boundary and produce a fine-grained mask.

- CTO Lin et al. (2023a) integrates Convolutional Neural Networks (CNNs), Vision Transformer (ViT), and a boundary detection operator (e.g., Sobel Kanopoulos et al. (1988)). The CNNs and ViT form the encoder, capturing feature dependencies, while the decoder combines convolutional layers and the boundary detection operator to enhance boundary segmentation. Specifically, a convolutional layer adaptively fuses the initial features from the boundary detection operator (Sobel operator) with the latent representations from the encoder for boundary refinement. Ground truth boundary maps guide and supervise this boundary learning process.

- Boundary-aware CNNs Hatamizadeh et al. (2019) utilize a standard encoder-decoder architecture alongside a shape processing component to process feature maps at the boundary level. The shape processing component incorporates an attention layer and a dilated spatial pyramid pooling layer to jointly learn boundary information, supervised by ground truth boundary maps that distinguish boundary and non-boundary pixels.

Both CTO Lin et al. (2023a) and Boundary-aware CNNs Hatamizadeh et al. (2019) require additional boundary maps for supervision, making them unsuitable for weakly supervised semantic segmentation. BoundaryCAM Prabakaran et al. (2023) would require adaptation for such tasks.

**Discussion about the third challenge (similar structures):** The presence of consistent, similar structures within specific organs or adjacent tissues is crucial for effective segmentation. These structures require models to capture localized information rather than relying solely on global features. If not adequately exploited, such similarities can hinder models' ability to distinguish regions accurately, especially in cases involving subtle or small lesions. For example, small lesions may be misclassified as healthy tissues, limiting models' learning capacity. Conversely, leveraging structural similarities can provide essential physiological and structural information for differentiating lesions from healthy tissues.

## O    DISCUSSION ABOUT AVERAGE BUFFER AND SIMILAR TECHNIQUES

To distinguish our devised Average Buffer from similar techniques (e.g., traditional prototypes and memory banks), we discuss their concepts and functionalities as follows:

- **Average Buffer:** This approach computes the average representation of a batch of data within a specific label during the model's data update process. It focuses on capturing generalized representations across samples, which are then used as reference samples for subsequent processing.

- **Traditional prototypes:** These typically represent the central or most representative samples of a label, derived by aggregating features or samples within the label. Prototypes aim to encapsulate the core characteristics of a category, aiding the model in distinguishing between clustering centers of different categories and consolidating similar representations within the same category.

- **Memory banks:** These are structures designed to store and manage large amounts of information, such as features or historical data learned by the model during training. Memory banks are dynamically updated or replaced throughout the training process, improving training efficiency and representational capacity.

In summary, while our designed Average Buffer shares similarities with traditional prototypes and memory banks, it differs by emphasizing the computation of average representations and their dynamic updates, which are central to its functionality.

