# OpenReview forum: "Dancing with Discrepancies: Commonality Specificity Attention GAN for Weakly Supervised Medical Lesion Segmentation"
_ICLR.cc/2025/Conference — Submitted to ICLR 2025_

### Official Review · Reviewer_EB1C · 2024-10-28

**Soundness:** 2
**Presentation:** 2
**Contribution:** 2
**Rating:** 5
**Confidence:** 4

**Summary:**

This paper introduces a novel approach to weakly-supervised medical image segmentation that integrates C-conv for intra-image discrepancy learning, effectively reducing boundary uncertainty. Additionally, it employs CSA mechanisms for inter-image discrepancy learning. The proposed method demonstrates state-of-the-art performance across three public benchmarks.

**Strengths:**

The idea of utilizing two convolutional layers with different receptive fields to enhance boundary detection is intriguing. The improvement over baseline models is substantial.

**Weaknesses:**

1. The discussion of motivation lacks depth. Three major challenges underpin this method:
a.	The intensity distribution of pathological images differs from that of healthy images, allowing classification networks to shortcut the learning process and overlook detailed spatial information.
b.	Lesion boundaries often appear ambiguous.
c.	Images frequently share similar anatomical structures.
Regarding the first challenge, most generative method-based approaches effectively address this issue [1-4]. For the second challenge, numerous studies have integrated boundary-aware modules into medical image segmentation [5-7], yet the authors do not discussion about existing literature. As for the third challenge, it is unclear why it is categorized as a challenge in the context of this work.
2. As a GAN-based method, the authors primarily discuss and compare their approach with CAM-based methods, neglecting comparisons with other GAN-based or diffusion-based techniques. Additionally, the domain-specific baselines referenced in the paper appear somewhat outdated.
3. The paper is not easy to follow. Especially the method part, which is difficult to understand and contains numerous ambiguities and unclear points (refer to the questions for specifics).

[1]. Hu, Xinrong, et al. "Conditional diffusion models for weakly supervised medical image segmentation." International Conference on Medical Image Computing and Computer-Assisted Intervention. Cham: Springer Nature Switzerland, 2023.

[2]. Li, Jinpeng, et al. "Fast non-markovian diffusion model for weakly supervised anomaly detection in brain mr images." International Conference on Medical Image Computing and Computer-Assisted Intervention. Cham: Springer Nature Switzerland, 2023.

[3].Cycles with Masked Conditional Diffusion for Unsupervised Anomaly Segmentation in MRI." International Conference on Medical Image Computing and Computer-Assisted Intervention. Cham: Springer Nature Switzerland, 2023.

[4]. Gonzalez-Jimenez, Alvaro, et al. "SANO: Score-based Diffusion Model for Anomaly Localization in Dermatology." Proceedings of the IEEE/CVF Conference on Computer Vision and Pattern Recognition. 2023.

 [5]. Prabakaran, Bharath Srinivas, Erik Ostrowski, and Muhammad Shafique. "BoundaryCAM: A Boundary-based Refinement Framework for Weakly Supervised Semantic Segmentation of Medical Images."

[6]. Lin, Yi, et al. "Rethinking boundary detection in deep learning models for medical image segmentation." International Conference on Information Processing in Medical Imaging. Cham: Springer Nature Switzerland, 2023.

[7]. Hatamizadeh, Ali, Demetri Terzopoulos, and Andriy Myronenko. "Boundary aware networks for medical image segmentation." arXiv preprint arXiv:1908.08071 10 (2019).

**Questions:**

1.	From my understanding, C-Conv detects the boundary and subsequently removes the local representation at that boundary. Could this lead to a loss of valuable information? Additionally, might this approach impact boundaries of certain sturctures within the foreground or background, not just the boundary between the foreground and background?
2.	In 272, what is the size of the reference samples and how they are selected and dynamically replaced?
3.	What distinguishes the proposed average buffer from traditional prototypes or memory banks?
4.	It seems that the generator only produces latent representations of the healthy distribution. How are the segmentation mask and the transformed healthy modality in Figure 6 generated?

---

> ### Author Response · Authors · 2024-11-22
> **Response to Reviewer EB1C (1/4)**
>
> We appreciate your constructive feedback, particularly the valuable references you suggested, which are closely related to our work. Below, we provide detailed responses to each of your suggestions and questions.
>
> >**W1.** The discussion of motivation lacks depth. Three major challenges underpin this method: a. The intensity distribution of pathological images differs from that of healthy images, allowing classification networks to shortcut the learning process and overlook detailed spatial information. b. Lesion boundaries often appear ambiguous. c. Images frequently share similar anatomical structures. Regarding the first challenge, most generative method-based approaches effectively address this issue [1-4]. For the second challenge, numerous studies have integrated boundary-aware modules into medical image segmentation [5-7], yet the authors do not discussion about existing literature. As for the third challenge, it is unclear why it is categorized as a challenge in the context of this work.
>
> We addressed your concerns regarding the three major challenges from the following perspectives:
>
> **Discussion on existing literature for Challenge a:**
> Regarding the first challenge, our solution aligns with your understanding. Most generative method-based approaches effectively address inherent distribution discrepancies, as described in the manuscript (Section 3.1: Motivation & Overview). Specifically, we propose using the generative adversarial network (GAN) within the CoinGAN model to exploit such discrepancies.
> Additionally, diffusion models from your suggested references [1-4] present viable alternatives as backbones. We have included a comparison with the diffusion model in **W2** (i.e., CG-CDM) and expanded our discussions in the Appendix to incorporate the references [1-4] you provided.
>
> **Discussion on existing literature for Challenge b:** We appreciate the supplementary references you provided, including BoundaryCAM [5], CTO [6], and boundary-aware CNNs [7]. Below, we summarize the working principles of these studies with boundary-aware modules:
>
>
> - **BoundaryCAM [5]** employs an unsupervised clustering strategy to extract clusters of pixels, which assist in defining an initial boundary of the target object. Subsequently, BoundaryCAM combines Class Activation Mapping (CAM) with the Floodfill [8] algorithm to refine this initial boundary and produce a fine-grained mask.
>
>
> - **CTO [6]** integrates Convolutional Neural Networks (CNNs), Vision Transformer (ViT), and a boundary detection operator (e.g., Sobel [9]). The CNNs and ViT form the encoder, capturing feature dependencies, while the decoder combines convolutional layers and the boundary detection operator to enhance boundary segmentation. Specifically, a convolutional layer adaptively fuses the initial features from the boundary detection operator (Sobel operator) with the latent representations from the encoder for boundary refinement. Ground truth boundary maps guide and supervise this boundary learning process.
>
> - **Boundary-aware CNNs [7]** utilize a standard encoder-decoder architecture alongside a shape processing component to process feature maps at the boundary level. The shape processing component incorporates an attention layer and a dilated spatial pyramid pooling layer to jointly learn boundary information, supervised by ground truth boundary maps that distinguish boundary and non-boundary pixels.
>
> Both CTO [6] and Boundary-aware CNNs [7] require additional boundary maps for supervision, making them unsuitable for weakly supervised semantic segmentation. BoundaryCAM [5] would require adaptation for such tasks. To refine the related work, we will add these references in the updated manuscript and supplement the discussions in the Appendix.
>
> **Clarifications on Challenge c:** The presence of consistent, similar anatomical structures within specific organs or adjacent tissues is crucial for effective segmentation. These structures require models to capture localized information rather than relying solely on global features. If not adequately exploited, such similarities can hinder a model's ability to distinguish regions accurately, especially in cases involving subtle or small lesions. For example, small lesions may be misclassified as healthy tissue, limiting models' learning capacity. Conversely, leveraging anatomical similarities can provide essential physiological and structural information for differentiating lesions from healthy tissues.
>
>
> To improve clarity, we will revise the explanation of Challenge c in the updated manuscript and expand the discussions of Challenges a and b in the Appendix.
>
> >[8] Kenneth P Fishkin and Brian A Barsky. An analysis and algorithm for filling propagation. In Computer-generated images, pages 56–76. Springer, 1985.  \
> [9] Kanopoulos, N., Vasanthavada, N., Baker, R.L.: Design of an image edge detection
> filter using the Sobel operator. IEEE J. Solid-State Circ. 23(2), 358–367 (1988).

---

> ### Author Response · Authors · 2024-11-22
> **Response to Reviewer EB1C (2/4)**
>
> >**W2.** As a GAN-based method, the authors primarily discuss and compare their approach with CAM-based methods, neglecting comparisons with other GAN-based or diffusion-based techniques. In extensive experiments,  Additionally, the domain-specific baselines referenced in the paper appear somewhat outdated.
>
> We appreciate the insightful suggestion and have incorporated the domain-specific diffusion model CG-CDM, referenced in [1], as a new baseline for comparison. CG-CDM is specifically tailored for medical weakly supervised semantic segmentation, making it a suitable counterpart for our study. In reference [1], the BraTS dataset is used to assess model performance, with image-level labels for training, and the reported results of CG-CDM are retrieved from [1]. Additionally, we have evaluated our proposed CoinGAN model on the same BraTS dataset. The results are summarized below:
>
> | Methods| DSC(%)&uarr; | JC(%)&uarr;  |
> |----------|----------|----------|
> | CG-CDM | 56.3  | 45.0 |
> | CoinGAN  | 89.4 | 82.3 |
>
> The experimental results highlight the superior performance of CoinGAN, further validating the effectiveness of our proposal.
>
> >**W3.** The paper is not easy to follow. Especially the method part, which is difficult to understand and contains numerous ambiguities and unclear points (refer to the questions for specifics).
>
> We clarify the detailed working principles of CoinGAN from two perspectives:
>
> **Clarification of CoinGAN's key components:**
>
> 1. C-Conv Module for Intra-image Representation Discrepancies: The C-Conv module is designed to learn intra-image representation discrepancies. Specifically, the Edge Perception Zone (EPZ) convolution, with its wider receptive field, enables earlier detection of potential change regions. The Internal Feature Zone (IFZ) convolution, with a smaller receptive field, primarily focuses on extracting local internal features. The fine-grained perception discrepancies between the EPZ and IFZ convolution are instrumental in identifying boundaries with label changes. This dual approach hence effectively addresses the challenge of ambiguous boundaries in lesion segmentation.
>
> 2. CSA Mechanism for Cross-modality Representation Alignment:
> The Commonality-Specificity Attention (CSA) mechanism consists of two components:
> - The Commonality Attention (CA) mechanism highlights similarity representations between pathological and healthy modalities, enhancing shared features.
> - The Specificity Attention (SA) mechanism identifies discrepancy representations between these modalities, isolating distinguishing features.
> - By emphasizing shared similarities and reducing discrepancies, the CSA mechanism enables the conversion of pathological representations into healthy representations. This helps address the challenge posed by similar anatomical structures in medical images.
>
> 3. CycleGAN for Cross-modality Conversion: CycleGAN is employed to perform image-to-image conversion from pathological to healthy modalities. By modeling the distribution discrepancy between these modalities, CycleGAN generates a converted healthy image. The difference between the original pathological image and its corresponding converted healthy image is then used to derive the lesion segmentation mask.
>
> To improve clarity, we will revise the description of CoinGAN in the introduction (Section 1) and provide an updated overview in Section 3.1, highlighting the interconnections and relationships among its key components.
>
> **Clarification of technical details:**
>
> We will systematically address the technical concerns raised in the questions of your review, revising and expanding the relevant details throughout the manuscript to ensure clarity and comprehensiveness. These updates will be reflected in the corresponding sections of the revised paper.

---

> ### Author Response · Authors · 2024-11-22
> **Response to Reviewer EB1C (3/4)**
>
> >**Q1.** From my understanding, C-Conv detects the boundary and subsequently removes the local representation at that boundary. Could this lead to a loss of valuable information? Additionally, might this approach impact boundaries of certain sturctures within the foreground or background, not just the boundary between the foreground and background?
>
> We appreciate your question regarding the potential impact of the C-Conv module on information retention and boundary sensitivity.
>
> **Boundary between the foreground and background:** We acknowledge that the C-Conv module may occasionally result in a minor loss of information; however, this trade-off contributes to capturing more valuable boundary information for object segmentation. Specifically:
>
> - The boundary regions detected by the C-Conv module are reweighted and set to "0." This distinction between "boundary" and "non-boundary" regions enhances CoinGAN's ability to focus on meaningful boundary information.
>
> - As demonstrated in the ablation study (Table 6), integrating the C-Conv module significantly improves segmentation performance.
>   - The backbone alone achieves a DSC of 65.15% (Table 6, first row).
>   - Incorporating the C-Conv module increases the DSC to 68.29% (Table 6, second row).
> This improvement highlights the effectiveness of the C-Conv module in capturing boundary information, confirming that it primarily aids segmentation rather than causing substantial loss of valuable information.
>
> **Boundaries of certain structures within the foreground or background:** To address the possibility of the C-Conv module inadvertently impacting boundaries within foreground or background structures, we introduce the hyperparameter $\lambda$. This hyperparameter adjusts the sensitivity of the C-Conv module to boundary information, preventing it from becoming overly sensitive and mitigating the risk of losing critical object details. As shown in Figure 4, the model achieves optimal performance when $\lambda = 0.3$. This demonstrates that careful calibration of $\lambda$ balances the trade-off between boundary sensitivity and information retention, ensuring robust segmentation performance.
>
>
> In summary, the C-Conv module contributes positively to boundary detection for segmentation, and its design, complemented by the $\lambda$ hyperparameter, effectively minimizes potential drawbacks.
>
> >**Q2.** In 272, what is the size of the reference samples and how they are selected and dynamically replaced?
>
> In our experiments, the size of the reference samples is set to 4, aligning with the batch size. As the reference samples are primarily maintained in the Average Buffer, we detail its implementation below to clarify the dynamic replacement process for the reference samples.
>
> **Average Buffer**
>
> The Average Buffer is a sample buffer designed to store a certain number of reference samples and compute their average representations. These stored samples are dynamically updated with each batch of input samples. The formal mathematical definition is as follows:
>
> Let:
> - $\mathcal{A} = \\{ \mathbf{A}_1, \mathbf{A}_2, \dots, \mathbf{A}_N \\}$ represent the set of $N$ stored reference samples in the Average Buffer, where $N$ is the batch size.
> - $\mathbf{A}_i$ denote the $i$-th sample in the Average Buffer.
>
> The Average Buffer computes the average representation $\mathbf{\bar{A}}$ of the stored samples as:
>
>   $$\mathbf{\bar{A}} = \frac{1}{N} \sum _ {i=1}^{N} \mathbf{A} _ i$$
>
> where $\mathbf{A} _ i$ refers to the representation derived from the C-Conv module, specifically $\mathbf{C}\cdot \mathbf{F} _ {\text{IFZ}}(\mathbf{X} _ {h})$.
>
> During model training, when a new batch of samples $\\{ \mathbf{A}^{\prime} _ 1, \mathbf{A}^{\prime} _ 2, \dots, \mathbf{A}^{\prime} _ N \\}$ is input to the Average Buffer, the existing samples $\\{ \mathbf{A} _ 1, \mathbf{A} _ 2, \dots, \mathbf{A} _ N \\}$ are dynamically replaced by the new batch $\\{ \mathbf{A}^{\prime} _ 1, \mathbf{A}^{\prime} _ 2, \dots, \mathbf{A}^{\prime} _ N \\}$.
>
> After replacement, the updated Average Buffer is:
> $$
> \mathcal{A} _ {new} = \\{ \mathbf{A}^{\prime} _ 1, \mathbf{A}^{\prime} _ 2, \dots, \mathbf{A}^{\prime} _ N \\}
> $$
> and the average representation of the new Average Buffer becomes:
> $$
> \mathbf{\bar{A}} _ {new} = \frac{1}{N} \sum _ {i=1}^{N} \mathbf{A}^{\prime} _ i
> $$
> where $\mathbf{A}^\prime _ i$ is the representation computed by the C-Conv module for the new batch of samples.
>
> To clarify this process, we will provide updated technical details in the Appendix.

---

> ### Author Response · Authors · 2024-11-22
> **Response to Reviewer EB1C (4/4)**
>
> >**Q3.** What distinguishes the proposed average buffer from traditional prototypes or memory banks?
>
> To distinguish our devised Average Buffer from these two techniques, we clarify their underlying concepts as follows:
>
> - **Average Buffer:** This approach computes the average representation of a batch of data within a specific label during the model's data update process. It focuses on capturing generalized representations across samples, which are then used as reference samples for subsequent processing.
>
> - **Traditional prototypes:** These typically represent the central or most representative samples of a label, derived by aggregating features or samples within the label.
> Prototypes aim to encapsulate the core characteristics of a category, aiding the model in distinguishing between clustering centers of different categories and consolidating similar representations within the same category.
>
> - **Memory banks:** These are structures designed to store and manage large amounts of information, such as features or historical data learned by the model during training. Memory banks are dynamically updated or replaced throughout the training process, improving training efficiency and representational capacity.
>
> In summary, while our designed Average Buffer shares similarities with traditional prototypes and memory banks, it differs by emphasizing the computation of average representations and their dynamic updates, which are central to its functionality. We will further elaborate on these points in the Appendix.
>
> >**Q4.** It seems that the generator only produces latent representations of the healthy distribution. How are the segmentation mask and the transformed healthy modality in Figure 6 generated?
>
> We clarify the generation processes for converted healthy images, the segmentation mask, and the latent representations of the healthy distribution as follows:
>
> As described in our clarification of CoinGAN's key components in **W3**:
> - The generator is responsible for producing the converted healthy images.
> - The segmentation mask for the lesion region is derived from the difference between the original image and the converted healthy image.
> - The latent representations of the healthy distribution are located in the network layer immediately preceding the generator's final output layer.
>
> We will further detail these processes in the revised methodology section of the updated manuscript.

---

> > ### Comment · Reviewer_EB1C · 2024-11-25
> >
> > Thank you for your response. While some of my concerns have been addressed, I believe the main contribution of this work lies in specific techniques tailored to this framework, whose high-level ideas have been discussed in prior research. Furthermore, I don't see its potential for generalization to other tasks or for providing insights that would appeal to a wider audience. Additionally, advancements over GAN are no longer at the forefront of research, making it challenging to gauge significant impact. Therefore, I feel this paper is better suited for a domain-specific conference or journal rather than a general ML conference. I continue to lean towards a negative rating.

---

> > > ### Author Response · Authors · 2024-11-25
> > > **Follow up: the updated manuscript for Reviewer EB1C**
> > >
> > > Hi, Reviewer EB1C, thank you for your insightful comments and suggestions. We have thoroughly revised the manuscript point by point in response to your feedback, with the corresponding changes clearly highlighted in blue. The detailed updates are outlined below:
> > >
> > > **W1.**
> > > - We have revised the explanation of Challenge c in lines 092-095 and lines 202-206 of the updated manuscript.
> > > - We have expanded the discussions of Challenges a and b in Appendix N.
> > >
> > > **W2.**
> > > - We have added a comparison with a state-of-the-art domain-specific diffusion-based model in Appendix J.
> > >
> > > **W3.**
> > > - We will revise the description of CoinGAN in the introduction (Section 1) and provide an updated overview in Section 3.1, highlighting the interconnections and relationships among its key components.
> > > - More methodological details have been updated in Section 3 of the updated manuscript.
> > >
> > > **Q2.**
> > > - We have updated the technical details of the dynamic replacement process for the reference samples in Appendix C.
> > >
> > > **Q3.**
> > > - We have updated the discussion of the Average Buffer and two related techniques in Appendix O.
> > >
> > > **Q4.**
> > > - We have refined the method in Section 3 to clarify the segmentation mask, the transformed healthy modality, and latent representations of the healthy distribution.
> > >
> > > We are glad that our previous responses addressed your concerns. The corresponding revisions have been provided in the updated manuscript.
> > >
> > > We hope the latest responses can further clarify your remaining questions. If you have any further questions or concerns, we remain fully prepared and eager to engage in additional discussions.

---

> ### Author Response · Authors · 2024-11-25
> **Response to Comment by Reviewer EB1C**
>
> Thank you for your feedback.
>
> >**New Q1. I believe the main contribution of this work lies in specific techniques tailored to this framework, whose high-level ideas have been discussed in prior research. Furthermore, I don't see its potential for generalization to other tasks or for providing insights that would appeal to a wider audience.**
>
> We would like to clarify that our work does not focus on specific techniques tailored to this framework. Instead, the use of discrepancy information is integral to a variety of deep learning tasks. To provide further clarity, we will elaborate on this from three perspectives:
>
> **Discrepancies are essential to classification and further segmentation across diverse applications.** Our work focuses on the discrepancies between different labels of medical images, independent of specific datasets, applications, or distribution patterns. As emphasized in the title, these discrepancies are fundamental to classification and segmentation tasks across various applications. Moreover, our proposal is highly adaptable and can be extended to domains such as face recognition, person re-identification, and object tracking, which share similarities with our setting. This generalizability underscores the potential of our work to inspire future research in these broader domains.
>
> **In the targeted medical domain, we have extended our proposal to include a new application: modeling the progression of a specific disease, further showcasing the versatility of CoinGAN.** Specifically, we have applied it to model the progression of mediastinal lesions from mild to severe, capturing the gradual worsening of the condition. As noted in our global response, due to the constraints for displaying the visualized maps in rebuttal, we will update the detailed experimental results in the paper shortly.
>
> **We have thoroughly reviewed your suggested related studies and confirmed that CoinGAN demonstrates superior performance** As detailed in our response to **W2**, we have analyzed the mechanisms, strengths, and limitations of these related studies. Additionally, we have compared CoinGAN with a new state-of-the-art, domain-specific diffusion model, and the experimental results, as presented in **W2**, further validate the superior performance of CoinGAN.
>
> >**New Q2: Additionally, advancements over GAN are no longer at the forefront of research, making it challenging to gauge significant impact.**
>
> As illustrated in the response of **W1**, most generative method-based approaches are promising alternative backbone, e.g., diffusion models you suggested in references 1-4. In our work, the generative adversarial network (GAN) has presented state-of-the-art performance over other recent WSSS works, as demonstrated in Table 1-5. Meanwhile, the newly added extensive experiments comparing our method with the recent domain-specific diffusion model demonstrated our method shows superior performance as presented in the results of **W2**.

---

### Official Review · Reviewer_SjzH · 2024-11-01

**Soundness:** 2
**Presentation:** 2
**Contribution:** 2
**Rating:** 5
**Confidence:** 4

**Summary:**

CoinGAN is designed for weakly supervised semantic segmentation (WSSS) in medical imaging. Key to CoinGAN is its use of a new convolution technique, contrastive convolution (C-Conv), and a dual attention mechanism (commonality-specificity attention.

**Strengths:**

The paper introduces a new form of convolution that helps accentuate fine-grained perceptual discrepancies within activation sub-maps, aiding in better delineation of lesion boundaries. A dual attention mechanism is used to suppress similarities in structural backgrounds across images while highlighting unique lesion characteristics.

**Weaknesses:**

1. The paper lacks a discussion on its generalizability across diverse medical imaging modalities and less common diseases.
2. The paper should provide a more detailed comparison with existing weak supervision methods, particularly those that do not use GAN architectures.
3. The paper lacks a detailed error analysis that could help identify the specific conditions under which the model performs poorly.
4. In my opinion, CoinGAN's performance is not acceptable since many semi-supervised models [1] can achieve much better performance with limited annotation.
[1] Li, Z., Li, Y., Li, Q., Wang, P., Guo, D., Lu, L., ... & Hong, Q. (2023). Lvit: language meets vision transformer in medical image segmentation. IEEE transactions on medical imaging.

**Questions:**

1. Can the model adapt to other forms of medical imaging data, such as MRI or CT images?
2. Since the author claims to use image-level annotation, I can understand that the label of the COVID dataset is normal and abnormal. However, the author needs to explain the use of the MonuSeg annotation.
3. How is the model's robustness to inaccuracies and variability in image-level labels?

---

> ### Author Response · Authors · 2024-11-22
> **Response to Reviewer SjzH (1/4)**
>
> Thank you for your valuable and insightful comments, which have significantly improved the quality of our work. Below, we have addressed each of your questions and suggestions in detail and outlined the corresponding changes made to the paper.
>
> >**W1 & Q1.** \
> **W1.** The paper lacks a discussion on its generalizability across diverse medical imaging modalities and less common diseases. \
> **Q1.** Can the model adapt to other forms of medical imaging data, such as MRI or CT images?
>
> **Generalizability across diverse medical imaging modalities:** \
> To address the concern regarding generalizability, we clarify that our extensive experiments encompass a wide range of medical imaging modalities:
> - QaTa-COV19 is a pneumonia lesion segmentation dataset that comprises chest **X-ray** images, enabling us to evaluate CoinGAN's performance in radiological imaging tasks.
> - ISIC2018 is a skin lesion segmentation dataset featuring **dermatoscopic images**, which allows us to assess the model's efficacy in dermatological applications.
> - MoNuSeg is a nuclear segmentation dataset based on **histopathologic images**, providing an opportunity to evaluate CoinGAN's effectiveness in tissue segmentation tasks within pathology.
>
> Further details can be found in Section 4 (Datasets & Metrics) and Appendix A (Datasets).
>
> **Evaluation on other forms of medical imaging data, such as MRI or CT images & less common diseases:** \
> To investigate the performance of CoinGAN on other forms of medical imaging data and less common diseases, we have newly included the BraTS 2021 dataset (https://www.kaggle.com/datasets/dschettler8845/brats-2021-task1) in our experiments for the detection and segmentation of brain tumors. This dataset focuses on MRI images and includes diverse cases of brain tumors, such as gliomas and other less common tumor types.
>
> We evaluate CoinGAN on this dataset to demonstrate its generalizability to the MRI modality and its applicability to less common disease types such as gliomas. The experimental results are summarized as follows:
>
> | Methods| DSC(%)&uarr; | JC(%)&uarr;  | ASD&darr;  | ACC(%)&uarr;| SP(%)&uarr;|SE(%)&uarr;|
> |----------|----------|----------| ----------|----------|----------|----------|
> |SeCo| 69.68 | 61.76 | 2.04 | 98.38 | 99.24 | 38.39 |
> | FPR | 86.07 | 77.80 | 0.67 | 97.88 | 98.80 | 74.91 |
> | CoinGAN  | 89.41 | 82.30 | 0.47 | 98.56 | 99.63 | 72.14 |
>
> The experimental results show that CoinGAN achieves superior performance, surpassing state-of-the-art image-level weakly supervised semantic segmentation models, including FPR and SeCo. Notably, FPR and SeCo were previously the top-performing models across the three datasets, as detailed in Tables 1–3.

---

> ### Author Response · Authors · 2024-11-22
> **Response to Reviewer SjzH (2/4)**
>
> >**W2.** The paper should provide a more detailed comparison with existing weak supervision methods, particularly those that do not use GAN architectures.
>
> We clarify that our work already includes an extensive comparison with recent non-GAN-based weakly supervised semantic segmentation methods, such as SFC [2], SeCo [3], and DRS [4], as presented in Tables 1–3. Below, we summarize the key characteristics of these compared methods:
> - SFC [2] follows the standard architecture of a feature encoder and a classifier. By integrating Class Activation Mapping (CAM) with an Image Bank, SFC calibrates shared features in the classifier weights for both head and tail classes. This calibration effectively addresses class activation imbalances, enhancing the performance in weakly supervised semantic segmentation. The backbone used in SFC is ResNet101.
> - SeCo [3] employs a knowledge distillation architecture with dual teachers and a single student. It addresses the co-occurrence problem in both image and feature spaces by decomposing images into patches with labeled regions in the image space and by contrasting multiple granularities of image knowledge in the feature space. This dual approach improves the handling of co-occurrence issues, thereby enhancing semantic feature representation. SeCo utilizes ViT-B/16 as its backbone.
> - DRS [4] employs a sequential architecture consisting of a classification network, a refinement network, and a segmentation network.
> The classification network generates coarse class activation maps, which the refinement network enhances through discriminative region suppression to improve localization accuracy. These refined maps are then used as pseudo-labels for training the segmentation network, enabling the transition from image-level to pixel-level labels.
>
> >[2] Zhao, Xinqiao, Feilong Tang, Xiaoyang Wang, and Jimin Xiao. "Sfc: Shared feature calibration in weakly supervised semantic segmentation." In Proceedings of the AAAI Conference on Artificial Intelligence, vol. 38, no. 7, pp. 7525-7533. 2024. \
> >[3] Yang, Zhiwei, Kexue Fu, Minghong Duan, Linhao Qu, Shuo Wang, and Zhijian Song. "Separate and conquer: Decoupling co-occurrence via decomposition and representation for weakly supervised semantic segmentation." In Proceedings of the IEEE/CVF Conference on Computer Vision and Pattern Recognition, pp. 3606-3615. 2024. \
> >[4] Kim, Beomyoung, Sangeun Han, and Junmo Kim. "Discriminative region suppression for weakly-supervised semantic segmentation." In Proceedings of the AAAI Conference on Artificial Intelligence, vol. 35, no. 2, pp. 1754-1761. 2021.
>
> >**W3.** The paper lacks a detailed error analysis that could help identify the specific conditions under which the model performs poorly.
>
> We appreciate this constructive comment, which deepens the understanding of our work. Below, we address your concern by elaborating on the core concept of our model - the utilization of **discrepancy** information. The discrepancies between different labels of medical images are pivotal for distinguishing discriminative regions (i.e., regions of interest), forming the foundation of our proposed CoinGAN. CoinGAN leverages this discrepancy information to enhance the segmentation of target objects in medical images.
>
> However, discrepancy information can be affected by potentially "inaccurate" image-level labels, which are not uncommon in clinical practice. For instance, some individuals labeled as healthy controls may exhibit subtle abnormalities in their medical images that resemble patient lesions but have not progressed to a diagnosable disease stage. These "healthy" images may introduce ambiguity into the model's learning process,
> impairing the effective use of discrepancy information and, consequently, the overall model performance.
>
> Moreover, such data quality issues stemming from the complexity of real-world clinical scenarios can also affect the performance of other weakly supervised semantic segmentation methods, as evidenced by the robustness study in **Q3**. Addressing these challenges remains a critical area for future exploration.

---

> ### Author Response · Authors · 2024-11-22
> **Response to Reviewer SjzH (3/4)**
>
> >**W4.** In my opinion, CoinGAN's performance is not acceptable since many semi-supervised models [1] can achieve much better performance with limited annotation. [1] Li, Z., Li, Y., Li, Q., Wang, P., Guo, D., Lu, L., ... & Hong, Q. (2023). Lvit: language meets vision transformer in medical image segmentation. IEEE transactions on medical imaging.
>
> We appreciate your reference to other related studies. However, we would like to emphasize that our work specifically addresses weakly supervised semantic segmentation tasks, which fundamentally differ from semi-supervised methods in terms of the supervisory information they utilize. Comparing our proposed weakly supervised semantic segmentation model with semi-supervised methods is not entirely fair, as the latter relies on additional textual annotations that provide richer supervisory signals.
>
> In image-level weakly supervised semantic segmentation tasks, the supervisory information is strictly limited to image-level labels (e.g., "pneumonia" vs. "healthy") without incorporating any explicit details about lesion locations or counts, etc. Our results, as demonstrated in Tables 1–5, show that CoinGAN achieves state-of-the-art performance compared to recent weakly supervised semantic segmentation counterparts, which operate under the same constraints.
>
> By contrast, semi-supervised methods, such as the one you suggested, leverage detailed textual annotations aligned with segmentation tasks. These annotations provide comprehensive information, such as lesion presence, counts, and specific locations. For instance, in the semi-supervised model Lvit you suggested, annotations like the following are utilized (the detailed example extracted from the Lvit paper):
> - "Bilateral pulmonary infection, two infected areas, upper left lung and upper right lung" describes the presence of bilateral lung infection with two infection areas located in the upper left and upper right lungs, respectively.
>
> In conclusion, while semi-supervised methods benefit from enhanced supervision through textual annotations, weakly supervised methods like CoinGAN are designed to operate with limited information. Therefore, a direct comparison between the two paradigms is not entirely appropriate. Nonetheless, we believe the performance of CoinGAN is highly competitive within the scope of weakly supervised semantic segmentation, as evidenced by its results relative to comparable methods.

---

> ### Author Response · Authors · 2024-11-22
> **Response to Reviewer SjzH (4/4)**
>
> >**Q2.** Since the author claims to use image-level annotation, I can understand that the label of the COVID dataset is normal and abnormal. However, the author needs to explain the use of the MonuSeg annotation.
>
> We elaborate on our use of the MoNuSeg annotations as follows. Consistent with the approach outlined in [5], we processed the MoNuSeg dataset by applying morphological operations (specifically image erosion) and Gaussian filtering to extract background information from the images. The resulting preprocessed images, representing the background information, are then paired with the original images to serve as two distinct image-level annotations.
>
> >[5] Yang, Xilin, Bijie Bai, Yijie Zhang, Musa Aydin, Yuzhu Li, Sahan Yoruc Selcuk, Paloma Casteleiro Costa et al. "Virtual birefringence imaging and histological staining of amyloid deposits in label-free tissue using autofluorescence microscopy and deep learning." Nature Communications 15, no. 1 (2024): 7978.
>
> >**Q3.** How is the model's robustness to inaccuracies and variability in image-level labels?
>
> To assess the robustness of CoinGAN to inaccuracies and variability in image-level labels, we have designed and supplemented the following experiment. Specifically, following the protocol of existing robustness studies such as [6], we intentionally introduce inaccuracies in 10% of the image-level labels and evaluate the performance of CoinGAN alongside the best-performing weakly supervised semantic segmentation baseline method under these conditions.
>
> For this robustness study, we use the QaTa-COV19 dataset as the reference. The experimental results, including a comparison of CoinGAN with the baseline, are summarized in the table below:
>
> | Methods| DSC(%)&uarr; | JC(%)&uarr;  | ASD&darr;  | ACC(%)&uarr;|
> |----------|----------|----------| ----------|----------|
> | FPR-10% | 63.20 (-1.62) | 48.99 (-2.54) | 2.70 (-0.28) | 72.74 (-4.34) |
> | CoinGAN-10%  | 70.04 (-1.65) | 57.19 (-1.52) | 2.03 (-0.58) | 81.88 (-0.23) |
>
> The values on the left represent the models' performance with 10% inaccurate image-level labels, while the values in parentheses indicate performance changes relative to the scenario without label inaccuracies. A "-" signifies a decline in performance. Detailed results for the scenario without label inaccuracies are provided in Table 1 of the manuscript. Notably, FPR is identified as the best-performing weakly supervised semantic segmentation model on the QaTa-COV19 dataset under image-level labels, as shown in Table 1.
>
> The experimental results reveal that both CoinGAN and the baseline method experience performance degradation across all four comprehensive metrics when subjected to inaccurate labels. However, CoinGAN exhibits smaller or comparable performance variations in three key metrics—DSC, JC, and ACC—compared to the baseline, highlighting its superior robustness against label inaccuracies.
>
> >[6] Wei, Hongxin, Lue Tao, Renchunzi Xie, and Bo An. "Open-set label noise can improve robustness against inherent label noise." Advances in Neural Information Processing Systems 34 (2021): 7978-7992.

---

> > ### Author Response · Authors · 2024-11-25
> > **Follow up: the updated manuscript for Reviewer SjzH**
> >
> > Hi, Reviewer SjzH, thank you for your valuable comments and suggestions. We have carefully revised the manuscript point by point based on your feedback, with the corresponding changes highlighted in blue. The detailed updates are as follows:
> >
> > **W1 & Q1.**
> > - We have refined the dataset description in lines 352-353 of our manuscript and supplemented more details in Appendix D.
> > - We have added the evaluation on other forms of medical imaging data and less common disease (BraTS2021) in Appendix I.
> >
> > **W3.**
> > - We have added a detailed error analysis in Appendix L.
> >
> > **Q2.**
> > - We have supplemented the details of the MoNuSeg annotations in Appendix E.
> >
> > **Q3.**
> > - We have added the robustness experiments to inaccuracies and variability in image-level labels in Appendix M.
> >
> > We hope the above responses and corresponding revisions in the manuscript effectively address your concerns. If you have any further questions, we are fully prepared and eager to engage in additional discussions.

---

> ### Comment · Reviewer_SjzH · 2024-11-26
>
> Thanks for the revision. However, the author did not address my concern. The author claims "Notably, FPR and SeCo were previously the top-performing models across the three datasets, as detailed in Tables 1–3.", I keep doubt about this claim. Additionally, the author cannot convince me why weak supervision learning is needed in medical image segmentation. Especially with only a small amount of annotations, it is possible to achieve a good performance. And CoinGAN can even not beat U-Net, which makes me doubt its effectiveness. Therefore, I will maintain my original rating and hold a negative vote for it.

---

> > ### Author Response · Authors · 2024-11-27
> > **Follow up: Response to Reviewer SjzH (1/3)**
> >
> > Thank you for your response.
> >
> > >**New Q1:** The author claims "Notably, FPR and SeCo were previously the top-performing models across the three datasets, as detailed in Tables 1–3.", I keep doubt about this claim.
> >
> > We would like to address this concern in two parts:
> >
> > - As introduced in Section 4.1, the comparison methods presented in Tables 1–3 are drawn from recent top-tier conference papers on weakly-supervised semantic segmentation, which is why we selected them for comparison.
> > - Specifically, the claim that "FPR and SeCo were previously the top-performing models across the three datasets" is supported by the experimental results presented in Tables 1–3. These tables summarize the performance of the comparison methods, where FPR and SeCo have achieved superior results compared to other methods on the three datasets. This claim is consistent with the experimental evidence.
> >
> > We hope this clarification addresses your concerns.

---

> > > ### Author Response · Authors · 2024-11-27
> > > **Follow up: Response to Reviewer SjzH (2/3)**
> > >
> > > >**New Q2:** Additionally, the author cannot convince me why weak supervision learning is needed in medical image segmentation. Especially with only a small amount of annotations, it is possible to achieve a good performance.
> > >
> > > >**why weak supervision learning is needed in medical image segmentation.**
> > >
> > > **The motivation of medical weakly supervised semantic segmentation:** The motivation for our research aligns with recent works on weakly-supervised semantic segmentation [7-10]. Specifically, semantic segmentation traditionally relies on pixel-level annotations, which require substantial human labor and time. In contrast, image-level weak supervision annotations are easier and less resource-intensive to obtain, alleviating the burden of data annotation. Among these forms of annotations, image-level labels are the most economical but also the most challenging to work with for segmentation tasks. This is because image-level labels only indicate the presence of an object without providing detailed spatial information, making the segmentation task more complex. Therefore, leveraging these weak annotations for training models presents a promising approach to reducing the annotation cost.
> > >
> > > Moreover, references [11-13] emphasize that medical image annotation requires specialized medical knowledge, further highlighting the urgency of weakly-supervised semantic segmentation in the medical field.
> > >
> > > >[7] Kweon, Hyeokjun, Sung-Hoon Yoon, and Kuk-Jin Yoon. "Weakly supervised semantic segmentation via adversarial learning of classifier and reconstructor." In Proceedings of the IEEE/CVF Conference on Computer Vision and Pattern Recognition, pp. 11329-11339. 2023. \
> > > >[8] Yang, Zhiwei, Kexue Fu, Minghong Duan, Linhao Qu, Shuo Wang, and Zhijian Song. "Separate and conquer: Decoupling co-occurrence via decomposition and representation for weakly supervised semantic segmentation." In Proceedings of the IEEE/CVF Conference on Computer Vision and Pattern Recognition, pp. 3606-3615. 2024. \
> > > >[9] Chen, Tao, Yazhou Yao, Xingguo Huang, Zechao Li, Liqiang Nie, and Jinhui Tang. "Spatial Structure Constraints for Weakly Supervised Semantic Segmentation." IEEE Transactions on Image Processing (2024). \
> > > >[10] Zhao, Xinqiao, Ziqian Yang, Tianhong Dai, Bingfeng Zhang, and Jimin Xiao. "PSDPM: Prototype-based Secondary Discriminative Pixels Mining for Weakly Supervised Semantic Segmentation." In Proceedings of the IEEE/CVF Conference on Computer Vision and Pattern Recognition, pp. 3437-3446. 2024. \
> > > >[11] Zhong, Yuan, Chenhui Tang, Yumeng Yang, Ruoxi Qi, Kang Zhou, Yuqi Gong, Pheng Ann Heng, Janet H. Hsiao, and Qi Dou. "Weakly-Supervised Medical Image Segmentation with Gaze Annotations." In International Conference on Medical Image Computing and Computer-Assisted Intervention, pp. 530-540. Cham: Springer Nature Switzerland, 2024. \
> > > >[12] Du, Hao, Qihua Dong, Yan Xu, and Jing Liao. "Weakly-supervised 3D medical image segmentation using geometric prior and contrastive similarity." IEEE Transactions on Medical Imaging 42, no. 10 (2023): 2936-2947. \
> > > >[13] Chen, Zhang, et al. "C-cam: Causal cam for weakly supervised semantic segmentation on medical image." Proceedings of the IEEE/CVF Conference on Computer Vision and Pattern Recognition. 2022.
> > >
> > > >**Especially with only a small amount of annotations, it is possible to achieve a good performance.**
> > >
> > > This question aligns with the **W4** you proposed. We would like to clarify that the semi-supervised Lvit model you proposed achieves good performance not merely by relying on a small amount of annotations, but rather by leveraging both pixel-level annotations and rich textual annotations, as presented in Table II of the Lvit paper.
> > >
> > > Specifically, we clarify it by analyzing the use of the annotations in the Lvit model. As in Table II of the Lvit paper, when pixel-level annotations are available, Lvit directly uses the pixel-level annotations for the model training. Specifically, Lvit provides three settings with varying proportions of pixel-level annotations (25%, 50%, or 100%) for the model training. Besides, Lvit leverages textual annotations with rich semantic information to supervise the model throughout the training process. These textual annotations include key details such as the presence of lesions, the number of lesions, and the locations of lesions. In contrast, our CoinGAN method uses only image-level annotations that only indicate the presence of a lesion, which significantly limits the available information. Therefore, the semi-supervised Lvit model you proposed achieves good performance not merely by relying on a small amount of annotations, but rather by leveraging both pixel-level annotations and rich textual annotations. Comparing our method directly with semi-supervised methods like Lvit, which relies on much more detailed supervision, would be unfair.

---

> > > > ### Author Response · Authors · 2024-11-27
> > > > **Follow up: Response to Reviewer SjzH (3/3)**
> > > >
> > > > >**New Q3:** And CoinGAN can even not beat U-Net, which makes me doubt its effectiveness.
> > > >
> > > > We would like to clarify this by referring to Table 1 in our paper.
> > > >
> > > > As reported in Table 1, U-Net is trained using pixel-level annotations for fully supervised learning, serving as a reference for evaluating the upper bound of the model performance on benchmark datasets. In contrast, our CoinGAN is designed to work with weakly-supervised annotations, specifically image-level labels. This comparison highlights that CoinGAN effectively performs in weakly-supervised settings, achieving performance that is closest to fully-supervised methods.
> > > >
> > > > This demonstrates the potential of CoinGAN to perform effectively with less detailed annotations.

---

### Official Review · Reviewer_LD6Q · 2024-11-03

**Soundness:** 2
**Presentation:** 1
**Contribution:** 2
**Rating:** 3
**Confidence:** 4

**Summary:**

The paper presents a method for weakly supervised medical lesion segmentation. The authors make two observations about the characteristics of lesion regions in the images and propose a GAN-based method that aims to exploit these characteristics to improve segmentation quality. The method is evaluated on three public benchmark datasets and compared with several state-of-the-art baselines. The paper also includes an ablation study.

**Strengths:**

* The observation-based approach to design the method is interesting. From what I understood from the observations, this might be an interesting direction of research.

* The evaluation is fairly extensive, with comparisons on multiple datasets and with a number of alternative methods.

* From the results, it seems that the proposed method outperforms the other methods in the experiments.

**Weaknesses:**

While the ideas behind the method could be interesting and the evaluation seems fairly extensive, I must admit that I found the paper very hard to follow.

From the Introduction, the assumptions and the general idea of what the method does remain unclear to me.

* Why do we need a GAN to learn from image-level labels? If we want to classify, detect, localize, segment something, why do we need a GAN? I don't think this is explained.

* The arguments about intensity and distributions are unclear. The terms are not defined (what exactly is a "sharp and high-intensity anatomical distribution" and how does this relate to the problem?). The assumptions also seem quite specific to these datasets and applications: does a high intensity always correlate with malignancies?

* The method apparently studies a "distribution shift" that is "driven" by a "GAN-based adversarial loss function", but from the Introduction it is unclear to me what this distribution shift indicates, and how it would benefit a weakly supervised segmentation model.

The description of the method is very technical and, at least for me, did not help to clarify what the method is intended to do and how it works.

Combined with the writing and word choice, which is often vague and imprecise, I found the presentation of the paper insufficient. There may be interesting ideas in the method -- apparently, it does improve performance -- but the paper did not help me to understand what they are and how they work.

**Questions:**

Some suggestions for improvement, highlighting some of the parts that I found unclear:

* Page #1 (Introduction):
  > a diverse array of computer vision tasks, e.g., autonomous driving Jiang et al. (2024), robotics Panda et al. (2023) and medical diagnosis Huang et al. (2024).

  These are oddly specific references for such a general statement.

* Page #1 (Introduction):
  > On the contrary, some weak supervision alternatives, e.g., image-level labels He et al. (2024), points Gao et al. (2024), and bounding boxes Cheng et al. (2023), are effortless to obtain.

  I understand they are cheaper/easier to obtain, but they are not "effortless".

* Page #1 (Introduction):
  > Image-level WSSS is extremely challenging since these image-level labels solely indicate the presence or absence of the target object without specifying any location information.

  Doesn't that also depend on the type of label? It could be the size of the object, or the severity, for example. It doesn't have to a binary present/not present.

* Page #2 (Introduction):
  > Our insight is that medical segmentation hinges on pronounced discernible information, image-level supervision is vulnerable to some medical challenges pointing to an unstable convergence but the inherent discrepancy information encapsulated within the images can assist in further diving into the whole discriminative regions.

  I have no idea what this sentence is meant to say, or what the subfigures on the left are supposed to show.

* Page #2 (Introduction):
  > but such models may not grasp what makes medical segmentation overflow and bad uncontrollable shape.

  This is grammatically incorrect, and I find it hard to understand what is meant here. What does "overflow" mean? And "bad uncontrollable shape" of what? Uncontrollable by whom?

* Page #2 (Introduction):
  > As in Figure 1 (Right), sharp regions (high-intensity distribution) typically indicate a lesion that deviates from normal tissues (homogeneous distribution). The anomalous distribution shifts (high → low) may excavate valuable knowledge gaps.

  I have no idea what this means. Is this supposed to say that high-intensity pixels always indicate disease? (That might hold for this application, but isn't true in a general sense.)

  What are "anomalous distribution shifts" and what does it mean that they "excavate" knowledge gaps?

* Page #2 (Introduction):
  > GAN

  Why do we need a GAN to learn from image level labels? Wasn't the goal to classify, detect, or localize something?

* Page #2 (Introduction):
  > by suppressing inter-image strong-related areas and accentuating weak-related areas.

  Related to what?

* Page #2 (Introduction):
  > The CSA mechanism is designed to explore inter-image structural anomalies

  What are "inter-image structural anomalies"?

* Page #2 (Introduction):
  > Finally, a GAN-based adversarial loss function drives the distribution shift.

  Why does the distribution shift need to be driven? What does that mean? And wouldn't we want to reduce a distribution shift?

* Page #4 (Motivation & Overview):
  > The second answer is that the output structure lacks the constraints of background information, that is, the ignorance of common knowledge makes a free boundary.

  This is quite vague. What "background knowledge" and how would this "common knowledge" prevent a "free boundary" (and what is that anyway)?

* Page #5 (Contrastive Convolution (C-Conv) Module):
  > Thus we propose a new form of convolution, C-Conv, to address the above ambiguous elements.

  What "elements" does this refer to? What is an "ambiguous element"?

* Page #5 (Commonality-Specificity Attention (CSA) Mechanism ):
  > the CSA mechanism is proposed to delve into the inter-image distribution discrepancies

  The verb "delve" is really vague: what does CSA mechanism do with the discrepancies? Does it try to reduce them? Does it make them stronger? Does it use them for something else?

---

> ### Author Response · Authors · 2024-11-22
> **Response to Reviewer LD6Q (1/8)**
>
> We thank the reviewer for the valuable comments and detailed suggestions. Below, we provide individual responses to each question and suggestion.
>
> >**W1.** While the ideas behind the method could be interesting and the evaluation seems fairly extensive, I must admit that I found the paper very hard to follow.
>
> We appreciate your recognition of our ideas and the thorough evaluation of our work. In response to the identified weaknesses and questions, we have carefully revised the full paper as suggested, paying special attention to the **Introduction** and **Methodology** sections.
> The corresponding changes will be included in the updated manuscript.
>
> >**W2.** From the Introduction, the assumptions and the general idea of what the method does remain unclear to me. \
> > **W2-1 & Q7.** \
> >**W2-1.** Why do we need a GAN to learn from image-level labels? If we want to classify, detect, localize, segment something, why do we need a GAN? I don't think this is explained.  \
> >**Q7.** Page #2 (Introduction):
> GAN  \
> Why do we need a GAN to learn from image-level labels? Wasn't the goal to classify, detect, or localize something?
>
> We address this question by elaborating on the motivation and methodology of our proposed CoinGAN.
>
> **Motivation:** As illustrated in Figure 1, significant distribution discrepancies exist between different labels of medical images, such as pathological and healthy modalities. These discrepancies contain crucial discriminative information, enabling differentiation between labels. Motivated by this observation, we aim to exploit label-based medical image conversion to capture discriminative information, such as lesions. For this image conversion task, Generative Adversarial Networks (GANs) provide a well-established framework.
>
>
> **Methodology:** For medical images with different labels (e.g., pathological and healthy), the C-Conv module first learns intra-image representation discrepancies. Specifically, the Edge Perception Zone (EPZ) convolution and the Internal Feature Zone (IFZ) convolution capture fine-grained discrepancies in boundary regions (with label changes), addressing ambiguous boundaries in lesion segmentation.
> Next, the CSA mechanism employs a commonality attention (CA) mechanism to capture similarity representations between pathological and healthy modalities and a specificity attention (SA) mechanism to capture discrepancy representations between these modalities. By enhancing similarity representations and suppressing discrepancy representations, the CSA mechanism facilitates the conversion from pathological to healthy representations, reducing interference from similar anatomical structures during training. Finally, the GAN performs the conversion from pathological to healthy images, capturing the distribution discrepancies between the two modalities. The difference between the original pathological image and the converted healthy image forms the segmentation mask for the lesion region.
>
> To the best of our knowledge, CoinGAN is the first model to leverage distribution conversion for weakly supervised semantic segmentation in medical imaging.
>
> Besides, to clarify the motivation and pipeline of our methodology, we will revise the introduction and Section 3.1 (Motivation & Overview) in the updated manuscript.

---

> ### Author Response · Authors · 2024-11-22
> **Response to Reviewer LD6Q (2/8)**
>
> >**W2-2.** The arguments about intensity and distributions are unclear. The terms are not defined (what exactly is a "sharp and high-intensity anatomical distribution" and how does this relate to the problem?). The assumptions also seem quite specific to these datasets and applications: does a high intensity always correlate with malignancies?
>
> We address the above question using Figure 1 to provide clarification.
>
> **Definitions of intensity and distribution:**
>
> Figure 1 (first row) illustrates the probability mass function (PMF) for two different labels of medical images. The horizontal axis represents the intensity values of image pixels (ranging from 0 to 255), while the vertical axis denotes the probability of each intensity value. Specifically:
> - *Intensity* refers to the numerical value of a pixel's intensity.
> - *Distribution* represents the PMF of intensity values in medical images.
>
> **"Sharp and high-intensity anatomical distribution" and its relationship to the problem:**
>
> As shown in Figure 1, pathological images often exhibit a distinct, sharp probability distribution concentrated within certain high-intensity value ranges. This pattern differs significantly from the distributions observed in healthy images, suggesting that these discrepancies are critical for distinguishing between labels. CoinGAN leverages this observation by converting images between different labels and segmenting lesion areas through its model design to capture these underlying discrepancies.
>
> **Clarification on the assumption about high intensity and malignancies:**
>
> We do not assume that high intensity universally correlates with malignancies. Instead, our approach emphasizes the discrepancies between different labels of medical images, independent of specific datasets, applications, or distribution patterns. CoinGAN focuses on capturing the underlying discrepancies that distinguish different labels, rather than directly associating high intensity with malignancies. By prioritizing label-based medical image conversion, CoinGAN generalizes to diverse datasets, applications, and distribution patterns, ensuring broader applicability beyond scenarios involving high-intensity malignancies.
>
>
> >**W2-3.** The method apparently studies a "distribution shift" that is "driven" by a "GAN-based adversarial loss function", but from the Introduction it is unclear to me what this distribution shift indicates, and how it would benefit a weakly supervised segmentation model.
>
> The term "distribution shift" highlights CoinGAN's central process of converting distributions between different labels of images. This process involves adaptively detecting and recalibrating similarity and discrepancy representations within different labels of images to achieve the conversion from one distribution pattern to another. In CoinGAN, this distribution conversion is collaboratively performed by the CSA mechanism and GAN. The detailed functionality and implementation of these components within the weakly supervised segmentation model have been discussed in **W2-1 & Q7**.
>
> We appreciate the reviewer for highlighting this issue, which helped us recognize the potential ambiguity in the use of this term. To enhance clarity, we have replaced "distribution shift" with "distribution conversion" throughout the manuscript.

---

> ### Author Response · Authors · 2024-11-22
> **Response to Reviewer LD6Q (3/8)**
>
> >**W3.** The description of the method is very technical and, at least for me, did not help to clarify what the method is intended to do and how it works.
>
> To address this concern and further clarify the functionality and working principles of our method, we will make the following revisions:
> - **Motivation & Overview** (Section 3.1): We will revise the motivation and overview of the proposed CoinGAN to offer a clearer understanding of its purpose and functionality. We hope this improves the overall clarity of the model's objectives and rationale.
> - **Key Components** (Sections 3.2-3.4): We will refine the descriptions of the three critical components, including the C-Conv module, CSA mechanism, and objective function. Additional technical details will be provided in the Appendix to aid understanding.
> - **Component Interconnections**: To enhance the logical flow of the paper, we will add explanations of the relationships between the components. These updates will ensure that readers can better understand how the components interact and contribute to the functionality of CoinGAN.
>
> We hope these updates address the concern and provide the necessary clarity regarding the structure and functionality of our method.
>
> >**W4.** Combined with the writing and word choice, which is often vague and imprecise, I found the presentation of the paper insufficient. There may be interesting ideas in the method -- apparently, it does improve performance -- but the paper did not help me to understand what they are and how they work.
>
> We will carefully revise the entire manuscript to improve its writing and word choice. Specifically, we will optimize the language and phrasing to ensure greater precision and clarity. Furthermore, we will address the weaknesses and questions you raised point by point, refining the presentation of our method to articulate its key ideas and functionality clearly. We hope these revisions more effectively convey the contributions of our work.

---

> ### Author Response · Authors · 2024-11-22
> **Response to Reviewer LD6Q (4/8)**
>
> >**Q1.** Page #1 (Introduction): a diverse array of computer vision tasks, e.g., autonomous driving Jiang et al. (2024), robotics Panda et al. (2023) and medical diagnosis Huang et al. (2024). \
> These are oddly specific references for such a general statement.
>
> We have removed the specific references to Jiang et al. (2024), Panda et al. (2023), and Huang et al. (2024) in the introduction, as per your suggestion. Instead, we have cited a comprehensive review by Mo et al. (2022) [1], which provides a general overview of state-of-the-art semantic segmentation technologies based on deep learning. This review covers a wide range of applications, including autonomous driving, robotics, and medical diagnosis.
>
> >[1] Mo, Yujian, Yan Wu, Xinneng Yang, Feilin Liu, and Yujun Liao. "Review the state-of-the-art technologies of semantic segmentation based on deep learning." Neurocomputing 493 (2022): 626-646.
>
> >**Q2.** Page #1 (Introduction):
> On the contrary, some weak supervision alternatives, e.g., image-level labels He et al. (2024), points Gao et al. (2024), and bounding boxes Cheng et al. (2023), are effortless to obtain. \
> I understand they are cheaper/easier to obtain, but they are not "effortless".
>
> We agree with your opinion that the term "effortless" may not accurately reflect the effort involved in obtaining weak supervision alternatives. To address this, we have revised the sentence to use "easier to obtain," which more appropriately conveys the relative simplicity and lower resource requirements of acquiring image-level labels, points, and bounding boxes. The revised sentence now reads:
>
> "On the contrary, some weak supervision alternatives, e.g., image-level labels He et al. (2024), points Gao et al. (2024), and bounding boxes Cheng et al. (2023), are easier to obtain."
>
> >**Q3.** Page #1 (Introduction):
> Image-level WSSS is extremely challenging since these image-level labels solely indicate the presence or absence of the target object without specifying any location information. \
> Doesn't that also depend on the type of label? It could be the size of the object, or the severity, for example. It doesn't have to a binary present/not present.
>
> We would like to clarify that, in the context of weakly supervised semantic segmentation, image-level labels typically refer to binary or multi-class labels that indicate the presence or absence of a specific object or category within an image, as outlined in the survey literature [2]. While other label types, such as object size or severity, may occasionally be considered in specialized applications, the predominant use in weak supervision settings remains focused on binary presence/absence labels.
> >[2] Chan, Lyndon, Mahdi S. Hosseini, and Konstantinos N. Plataniotis. "A comprehensive analysis of weakly-supervised semantic segmentation in different image domains." International Journal of Computer Vision 129, no. 2 (2021): 361-384.
>
> Additionally, we find your suggested application scenario compelling. To explore this, we have introduced a new dataset, MELA [3], designed to analyze the progression of mediastinal lesions. In this dataset, medical images are categorized based on lesion sizes, with smaller lesions labeled as mild and larger ones as severe. This labeling reflects object sizes rather than presence/absence. Using this dataset, we have conducted extensive experiments, including visualization studies, to investigate the conversion from mild to severe lesions, effectively simulating the progression of mediastinal lesions. This analysis offers significant clinical insights for disease prevention and management. The updated results and detailed discussions will be supplemented in the Appendix.
> >[3] Wang, Jun, Xiawei Ji, Mengmeng Zhao, Yaofeng Wen, Yunlang She, Jiajun Deng, Chang Chen, Dahong Qian, Hongbing Lu, and Deping Zhao. "Size‐adaptive mediastinal multilesion detection in chest CT images via deep learning and a benchmark dataset." Medical Physics 49, no. 11 (2022): 7222-7236.

---

> ### Author Response · Authors · 2024-11-22
> **Response to Reviewer LD6Q (5/8)**
>
> >**Q4.** Page #2 (Introduction):
> Our insight is that medical segmentation hinges on pronounced discernible information, image-level supervision is vulnerable to some medical challenges pointing to an unstable convergence but the inherent discrepancy information encapsulated within the images can assist in further diving into the whole discriminative regions. \
> I have no idea what this sentence is meant to say, or what the subfigures on the left are supposed to show.
>
> These sentences in the caption for Figure 1 aim to summarize two key insights of the paper, emphasizing the challenges in medical weakly supervised semantic segmentation that motivate our proposal.
>
> **Clarification of the sentence:**
>
> - "Medical segmentation hinges on pronounced discernible information," emphasizes that image-level weakly supervised medical segmentation relies on the discrepancies between different labels of images, which is the foundation of segmentation tasks in medical imaging.
> - "Image-level supervision is vulnerable to some medical challenges pointing to an unstable convergence," highlights that in medical weakly supervised semantic segmentation, image-level labels are insufficient to address key challenges illustrated in the left subfigure of Figure 1. These challenges include potential distribution discrepancies, ambiguous boundaries, and similar anatomical structures, which may result in unstable model training and hence inferior performance in semantic segmentation.
> - "But the inherent discrepancy information encapsulated within the images can assist in further diving into the whole discriminative regions," highlights the central idea of our proposal. It clarifies that the discrepancies within regions of the same image and between different labels of images enhance the segmentation of discriminative regions (i.e., regions of interest). This aligns with the roles of the C-Conv module and CSA mechanism in CoinGAN, respectively.
>
> To improve clarity, we will revise the caption of Figure 1 and the introduction in the updated manuscript.
>
> **Clarification of the subfigures:**
> - The left subfigure of Figure 1 (Page #2) visualizes challenges encountered in medical weakly supervised semantic segmentation, using coronavirus disease 2019 (COVID-19) as an example.
> - The right subfigure illustrates the core idea behind our proposal, addressing these challenges through discrepancy information for improved segmentation outcomes, which serves as the motivation for the design of CoinGAN.
>
> >**Q5.** Page #2 (Introduction):
> but such models may not grasp what makes medical segmentation overflow and bad uncontrollable shape. \
> This is grammatically incorrect, and I find it hard to understand what is meant here. What does "overflow" mean? And "bad uncontrollable shape" of what? Uncontrollable by whom?
>
> The referred sentence was intended to highlight that recent studies have not adequately addressed the underlying causes of oversegmentation and inaccurate segmentation shapes in medical image segmentation. Specifically:
> - **Oversegmentation** refers to instances where the segmentation results extend beyond the actual ground truth object region, incorrectly including background areas as part of the object (false positives). This results in a segmentation "overflow."
> - **Inaccurate shapes** describe cases where the segmented regions deviate significantly from the true shape of the target objects, failing to align with the ground truth lesions and leading to segmentation errors that misrepresent the actual lesion boundaries.
>
> We agree with your observation regarding the term "uncontrollable," which lacks precision. Accordingly, we have replaced "uncontrollable" with "inaccurate" and revised the sentence to improve clarity as follows:
>
> "but such models fail to consider the causes of oversegmentation and inaccurate shapes in medical segmentation."
>
> To provide further clarification, we have added detailed explanations of "oversegmentation" and "inaccurate shapes" in the Appendix.
>
> We hope these revisions effectively address the issues and improve the clarity of the intended meaning.

---

> ### Author Response · Authors · 2024-11-22
> **Response to Reviewer LD6Q (6/8)**
>
> >**Q6.** Page #2 (Introduction):
> As in Figure 1 (Right), sharp regions (high-intensity distribution) typically indicate a lesion that deviates from normal tissues (homogeneous distribution). The anomalous distribution shifts (high → low) may excavate valuable knowledge gaps. \
> **Q6-1.** I have no idea what this means. Is this supposed to say that high-intensity pixels always indicate disease? (That might hold for this application, but isn't true in a general sense.)
>
> **Clarification of high-intensity pixels:** We would like to clarify that we do not assume high-intensity pixels always correlate with malignancies. Figure 1 is presented solely as a visualized example to aid understanding and does not represent all possible application scenarios.
>
> **Applications and generality:** As mentioned in the response to **W2-2**, our proposed CoinGAN is designed with broad applicability and is not restricted to specific datasets or applications. For instance, as highlighted in **Q3**, it can analyze the progression of mediastinal lesions. CoinGAN focuses on identifying and leveraging discrepancies between different labels of images, emphasizing pattern conversion across these labels, which may encompass a variety of distribution patterns.
>
> CoinGAN is not confined to converting from high to low-intensity values but extends to capturing other variations in distribution patterns. These discrepancies provide valuable insights and explore the knowledge underlying different labels of medical images, serving as the foundation for distinguishing between image categories. This focus on leveraging distribution discrepancies is central to our methodology.
>
>
> >**Q6-2.**
> What are "anomalous distribution shifts" and what does it mean that they "excavate" knowledge gaps?
>
> We address this question in three parts:
> - **Generation of anomalous distributions**: Using the pneumonia lesion depicted in Figure 1 as an example, certain high-intensity pixels differ significantly from those in healthy images, representing anomalous distributions relative to the healthy modality. These anomalous distributions serve as indicators of pneumonia lesions. This explanation is consistent with our responses to **W2-2** and **Q6-1**.
> - **Distribution conversion:** As detailed in **W2-3**, the term "distribution shift" has been updated to "distribution conversion" to avoid ambiguity. This process refers to converting one distribution pattern to another by adaptively learning and adjusting the anomalous distributions. Through this conversion, images associated with one label are transformed into those of another label, enabling the model to leverage label-based distribution discrepancies effectively.
> - **Exploring knowledge underlying different labels:** The difference between the converted image and the original image provides discriminative information across labels, often corresponding to clinically significant objects of interest. To improve clarity, we have replaced the term "excavate" with "explore" and revised "knowledge gap" to "the knowledge underlying different labels of medical images." These updates ensure consistency and better convey the intended meaning.
>
> We have incorporated these revisions into the manuscript to enhance precision and clarity.
>
> >**Q8.** Page #2 (Introduction):
> by suppressing inter-image strong-related areas and accentuating weak-related areas. \
> Related to what?
>
> The referred phrase describes the core functionality of the CSA mechanism. As explained in our responses to **W2-1 & Q7**, the CSA mechanism is designed to identify and utilize both the similarities and discrepancies between different labels of medical images (e.g., pathological and healthy images). Specifically:
> - "Related" areas refer to the associations between one label of medical images and another.
> - We have revised "strong-related areas" to "similarity regions between different labels of images", which emphasizes the regions where the labels share similar characteristics.
> - We have revised "weak-related areas" to "discrepancy regions between different labels of images", highlighting the areas where significant differences between the labels are observed.
>
> >**Q9.** Page #2 (Introduction):
> The CSA mechanism is designed to explore inter-image structural anomalies \
> What are "inter-image structural anomalies"?
>
> As explained in our response to **Q8**, the CSA mechanism is primarily designed to explore the similarities and discrepancies between different labels of medical images. In this context, "inter-image structural anomalies" refers to the structural discrepancies between images of different labels. These discrepancies serve as critical discriminative information for distinguishing between labels, forming the foundation of medical image segmentation, as also highlighted in **Q4**.

---

> ### Author Response · Authors · 2024-11-22
> **Response to Reviewer LD6Q (7/8)**
>
> >**Q10.** Page #2 (Introduction):
> Finally, a GAN-based adversarial loss function drives the distribution shift. \
> Why does the distribution shift need to be driven? What does that mean? And wouldn't we want to reduce a distribution shift?
>
> As detailed in our responses to **W2-1 & Q7**, the distribution discrepancies between different labels of images serve as the primary motivation for our work. To effectively capture these discrepancies, we employ a process referred to as "distribution conversion." As clarified in **W2-3**, distribution conversion involves transforming one distribution pattern into another by adaptively adjusting the similarity representations and discrepancy representations between different labels of images.
>
> In CoinGAN, the CSA mechanism and GAN collaboratively facilitate this distribution conversion, ensuring that the model captures meaningful distribution differences between labels. To avoid ambiguity, we have replaced the term "distribution shift" with "distribution conversion" in **W2-3**. This refined terminology better reflects the process of uncovering valuable information to distinguish between different labels of images.
>
> >**Q11.** Page #4 (Motivation & Overview):
> The second answer is that the output structure lacks the constraints of background information, that is, the ignorance of common knowledge makes a free boundary.  \
> This is quite vague. What "background knowledge" and how would this "common knowledge" prevent a "free boundary" (and what is that anyway)?
>
> To clarify, we address this question from two perspectives:
> - **Clarification of concepts:**
>   - "Background knowledge" refers to the similar regions between different labels of medical images, essentially representing the background regions.
>   - The discrepant regions between different labels of images constitute the foreground.
>   - The term "free boundary" refers to inaccurate segmentation shapes, where the segmentation boundary deviates from the true object boundary.
> - **Mechanism**:
>   As elaborated in **W2-1 & Q7**, the CSA mechanism employs:
>   - A commonality attention (CA) mechanism to capture similarity representations between different labels of medical images (e.g., pathological and healthy modalities).
>   - A specificity attention (SA) mechanism to capture discrepancy representations between labels.
>
>   During this process:
>   - The removal of discrepancy representations facilitates the conversion of the current image's distribution pattern into another distribution pattern.
>   - The addition of similarity representations enhances the background information in the detected image.
>
> This enhancement of background information enables CoinGAN to constrain the boundaries of the foreground object, thereby improving segmentation accuracy and addressing the issue of inaccurate segmentation shapes (previously referred to as "free boundary").
>
> To improve clarity, we have revised the phrase "free boundary" to "inaccurate segmentation shapes" in the updated manuscript.

---

> ### Author Response · Authors · 2024-11-22
> **Response to Reviewer LD6Q (8/8)**
>
> >**Q12.** Page #5 (Contrastive Convolution (C-Conv) Module):
> Thus we propose a new form of convolution, C-Conv, to address the above ambiguous elements. \
> What "elements" does this refer to? What is an "ambiguous element"?
>
> We would like to clarify these concepts from two perspectives:
> - **Conceptual explanation:** The term "element" refers to a representation computed by the convolutional operation at a specific position in the image. This representation summarizes the local information from the surrounding region at the corresponding position. "Ambiguous elements" generally describe representation values at boundary regions that are challenging to distinguish. To improve clarity, we have replaced the term "element" with "representation" and refined the descriptions for better expression.
>
> - **Causes of ambiguous representations:** Convolution is an operation where a convolutional kernel (a weight matrix) slides across the input image to extract local features progressively. For each position in the image, the convolution operation calculates the sum of the products between the kernel weights and the corresponding pixel values within the receptive field. Boundary regions, typically located at the intersection of the target foreground and the structural background, contain mixed information from both. As a result, the convolution outcomes in these regions often reflect features from both foreground and background, leading to ambiguity in the representations. This phenomenon is the root cause of ambiguous representations.
>
> As part of our revisions, we have updated the term "ambiguous elements" to "ambiguous representations" throughout the manuscript for consistency and clarity.
>
> >**Q13.** Page #5 (Commonality-Specificity Attention (CSA) Mechanism ):
> the CSA mechanism is proposed to delve into the inter-image distribution discrepancies \
> The verb "delve" is really vague: what does CSA mechanism do with the discrepancies? Does it try to reduce them? Does it make them stronger? Does it use them for something else?
>
> As detailed in our response to **Q11** and elaborated in the paper (Pages #5-6), the CSA mechanism combines a commonality attention (CA) mechanism and a specificity attention (SA) mechanism.
> - The CA mechanism captures similarity representations between different labels of medical images (e.g., pathological and healthy modalities).
> - The SA mechanism captures discrepancy representations between different labels of medical images.
>
> During this process, the CSA mechanism removes discrepancy representations to convert the current image's distribution pattern into another distribution pattern, facilitating distribution conversion. Simultaneously, it enhances similarity representations to improve background information in the detected image. This enhancement constrains the boundaries of the foreground object, thereby improving segmentation accuracy.
>
> The primary goal of the CSA mechanism is to adjust the distribution pattern by minimizing discrepancies and enhancing commonalities, thereby enabling the GAN to effectively perform distribution conversion. Beyond these operations, the CSA mechanism does not utilize the identified discrepancies for other purposes. To address potential ambiguities, we have revised the use of the term "delve" in the updated manuscript for clarity.

---

> > ### Author Response · Authors · 2024-11-25
> > **Follow up: the updated manuscript for Reviewer LD6Q**
> >
> > Hi, Reviewer LD6Q, we have carefully revised the manuscript point by point in response to your comments and suggestions. The corresponding changes have been made and are highlighted in blue. The detailed updates are as follows:
> >
> > **W1.**
> > -  We have carefully revised the full paper as suggested, paying special attention to Section 1 (Introduction) and Section 3 (Methodology). The corresponding changes will be included in the updated manuscript.
> >
> > **W2.** \
> > **W2-1 & Q7.**
> > - Aligning with the clarification in the responses of **W2-1 & Q7**, we have revised the introduction and Section 3.1 (Motivation & Overview) in the updated manuscript to illustrate the motivation and the corresponding methodology.
> >
> > **W2-2.**
> > - We have revised definitions of intensity and distribution in Appendix B.
> > - To avoid ambiguity, we have removed certain case-specific expressions and have revised the method description in the Introduction (lines 097-109). The updated expressions
> >
> > **W2-3.**
> > - We have replaced "distribution shift" with "distribution conversion" throughout the manuscript.
> >
> >
> > **W3.**
> > - We have revised the motivation and overview of the proposed CoinGAN in Section 1 and Section 3.1 to offer a clearer understanding of its purpose and functionality.
> > - We have refined the descriptions of the three critical components, including the C-Conv module, CSA mechanism, and objective function, in Section 3.2-3.4.
> > - To improve the logical flow of the paper, we have integrated explanations of the relationships between the components, including the problem to be addressed and the corresponding solution. The corresponding changes are updated in Section 3.
> >
> > **W4.**
> > - We have carefully revised the entire manuscript to improve its writing and word choice.
> >
> > **Q1.**
> > - We have removed the specific references to Jiang et al. (2024), Panda et al. (2023), and Huang et al. (2024) in the introduction, as per your suggestion. Instead, we have cited a comprehensive review by Mo et al. (2022) [1], which provides a general overview of state-of-the-art semantic segmentation technologies based on deep learning. The corresponding changes are updated in line 040.
> >
> > **Q2.**
> > - We have revised the sentence to use "easier to obtain" in line 043.
> >
> > **Q3.**
> > - We have explored the progression of mediastinal lesions with labels reflecting severity levels rather than binary presence/absence in Appendix K. The corresponding dataset description is added in Appendix E.
> >
> > **Q4.**
> > - We have revised the caption of Figure 1 (lines 067-074) and the introduction (lines 080-083 and lines 097-110) in the updated manuscript, which aligns with our response in Q4.
> >
> > **Q5.**
> > - We have revised the sentence in lines 078-079 and added detailed explanations of "oversegmentation" and "inaccurate shapes" in Appendix A.
> >
> > **Q6.** \
> > **Q6-1.**
> > - Regarding the general applications, we have removed certain case-specific expressions and have focused on the method description in the Introduction (lines 097-109).
> >
> > **Q6-2.**
> > - We have revised "anomalous distribution shifts" to "discrepancy distribution".
> > - The term "distribution shift" has been updated to "distribution conversion" to avoid ambiguity.
> > - We have replaced the term "excavate" with "explore" and revised "knowledge gap" to "the knowledge underlying different labels of medical images."
> >
> > **Q8.**
> > - We have revised "strong-related areas" to "similarity regions between different labels of images" throughout the manuscript.
> > - We have revised "weak-related areas" to "discrepancy regions between different labels of images" throughout the manuscript.
> >
> > **Q9.**
> > - We have revised "inter-image structural anomalies" to "inter-image distribution discrepancies" throughout the manuscript.
> >
> > **Q10.**
> > - We have replaced "distribution shift" with "distribution conversion" throughout the manuscript as in **W2-3.**
> > - Additionally, we have refined the motivation in the Introduction (Section 1) and the overview in the Methodology (Section 3).
> >
> > **Q11.**
> > - We have revised the phrase "free boundary" to "inaccurate segmentation shapes" in line 206 of the updated manuscript.
> > - We have refined "The second answer" in lines 201-206.
> >
> > **Q12.**
> > -  We have replaced the term "element" with "representation" and refined the descriptions for better expression throughout the manuscript.
> > -  "ambiguous element" have been revised to "ambiguous representations".
> >
> > **Q13.**
> > - We have revised the description of the CSA mechanism in Section 3.3 for clarity.
> > - We have revised the use of the term "delve" throughout the manuscript.
> >
> >
> > We hope that the above responses and the corresponding changes in the manuscript address your concerns. If you have any further questions, we would be happy to engage in additional discussions.

---

> > > ### Comment · Reviewer_LD6Q · 2024-11-25
> > > **Response to authors' response**
> > >
> > > I would like to thank the authors for their elaborate responses.
> > >
> > > My main concern was the presentation of the method, and despite the changes and response from the authors I still find it confusing.
> > >
> > > * The motivation (why use a GAN to convert images, if what we actually want is a segmentation?). In the Abstract and Introduction, I still find it difficult to pinpoint the lines where the authors describe the main point of their method.
> > >
> > >   The response illustrates the problem, I think (part 1/8):
> > >
> > >   > *Motivation:* As illustrated in Figure 1, significant distribution discrepancies exist between different labels of medical images [...] For this image conversion task, Generative Adversarial Networks (GANs) provide a well-established framework.
> > >   >
> > >   > *Methodology:* For medical images with different labels (e.g., pathological and healthy), the C-Conv module first learns intra-image representation discrepancies. [...] The difference between the original pathological image and the converted healthy image forms the segmentation mask for the lesion region.
> > >
> > >   This, like the paper, is a long, detailed, low-level description of how the method works. How it actually solves the problem (segmentation!) doesn't become clear until the very last sentence.
> > >
> > >   In a similar way, the Abstract and Introduction mostly discuss "distribution discrepancies", "distribution conversion" et cetera. How it actually produces a segmentation is hardly addressed.
> > >
> > >   In the Abstract, I think the key point of how the method works is hidden in this sentence:
> > >
> > >   > Then a commonality-specificity attention mechanism is proposed to suppress similarity regions between different labels of images and accentuate discrepancy regions between different labels of images.
> > >
> > >   but this is a very roundabout way to describe segmentation.
> > >
> > > * The assumptions and generalizability.
> > >
> > >   Much of the paper discusses "high-intensity anatomical distributions". Asked about how this would generalize to other medical applications, where perhaps the intensity is less clearly related to a disease, the authors response states (2/8) that
> > >
> > >   > We do not assume that high intensity universally correlates with malignancies. [...] By prioritizing label-based medical image conversion, CoinGAN generalizes to diverse datasets, applications, and distribution patterns, ensuring broader applicability beyond scenarios involving high-intensity malignancies.
> > >
> > >   I find this somewhat unconvincing. Intensity and intensity distributions play an important role in the paper, but the generalization to other types of images is not discussed. The malignancies in the experiments seem mostly intensity-based as well.
> > >
> > >   (And I still don't know what an "anatomical distribution" is. "Anatomical" suggests a spatial component, but the plots suggest a simple pixel intensity distribution.)
> > >
> > > I thank the authors again for the improvements made to the manuscript, but I will maintain my overall rating.

---

> > > > ### Author Response · Authors · 2024-12-03
> > > > **Further response to Reviewer LD6Q (1/2)**
> > > >
> > > > >**New Q1**. The motivation (why use a GAN to convert images, if what we actually want is a segmentation?). In the Abstract and Introduction, I still find it difficult to pinpoint the lines where the authors describe the main point of their method.
> > > >
> > > > **Background:** As outlined in our response of **W2-1 & Q7**, pronounced distribution discrepancies between different labels of medical images contain crucial discriminative information, enabling differentiation between labels. To leverage these distribution discrepancies for the object segmentation, we perform the label-based medical image conversion by the typical generative adversarial network (GAN).
> > > >
> > > > **Insights:** Specifically, the inherent discrepancies within regions of the same image and between different labels of images are used to enhance the segmentation of discriminative regions. The discrepancies between different objects within the same image can help distinguish different regions in one image while the discrepancies between different labels of medical images further assist the model in recognizing the key regions that can distinguish two images.
> > > > In this process, the intra-image and inter-image discrepancies are learned to strengthen the similarity representations between different labels of images and eliminate the discrepancy representations between different labels of images, facilitating a label-based image conversion from one label to another. The generator in the GAN framework can assist in this label-based image conversion, while the discriminator within GAN leverages image-level labels to supervise this image conversion.
> > > >
> > > > **Solution:** As in Figure 2, the C-Conv module is devised to explore the intra-image discrepancies, where fine-grained perceptual discrepancies of activation sub-maps within the same image adaptively reweight the intra-image boundary representations to ensure a clear distinction of different regions within images. Subsequently, the commonality-specificity attention (CSA) mechanism is proposed to recognize inter-image discrepancies. In this process, similarity representations between different labels of images are enhanced and discrepancy representations between different labels of images are filtered out. This steers the model's attention to the object regions while facilitating the image conversion from one label to another. Finally, representations enhanced by the C-Conv module and CSA mechanism are fed into a GAN network. The generator generates the converted image belonging to another label. The discriminator utilizes image-level annotations to supervise the intra-image and inter-image discrepancies learning. The discrepancies between the original image and the converted image are the segmentation masks that can distinguish different labels.
> > > >
> > > > More details can be seen in Section 3 of the updated manuscript (Methodology).

---

> > > > > ### Author Response · Authors · 2024-12-03
> > > > > **Further response to Reviewer LD6Q (2/2)**
> > > > >
> > > > > >**New Q2**. The assumptions and generalizability. \
> > > > > I find this somewhat unconvincing. Intensity and intensity distributions play an important role in the paper, but the generalization to other types of images is not discussed. The malignancies in the experiments seem mostly intensity-based as well.
> > > > >
> > > > > We clarify it through the following three perspectives:
> > > > >
> > > > > **The clarification of intensity:** In existing computer vision tasks, the images are fed into the devised models to extract the features from the images for various downstream tasks, where all pixel intensities are used to iteratively learn meaningful representations through multiple layers of linear and nonlinear transformations. Therefore, the pixel intensities are widely used across all visual tasks. \
> > > > > **The clarification of intensity distributions:** As discussed in the response of **New Q1**, there exist pronounced distribution discrepancies between different labels of medical images that contain crucial discriminative information. Inspired by this, we propose the C-Conv module to learn the representation discrepancies within the same image, where distribution discrepancies are inherently present when different objects exist within the same image. Furthermore, we introduce the CSA mechanism to learn the representation discrepancies between different labels of images, where these discrepancies naturally exist and serve to distinguish between labels. \
> > > > > **The generalization to other types of images:**
> > > > > - **Extensive experiments on different types of medical imaging data:** We have evaluated the effectiveness of our model on different types of medical imaging data, including X-rays (QaTa-COV19 dataset), dermatoscopic images (ISIC2018 dataset), and histopathologic images (MoNuSeg dataset). Additionally, to demonstrate the model's generalizability to other types of medical images, we conducted an extensive experiment on MRI images using the BraTS dataset. The experimental results are summarized as follows:
> > > > >
> > > > > | Methods| DSC(%)&uarr; | JC(%)&uarr;  | ASD&darr;  | ACC(%)&uarr;| SP(%)&uarr;|SE(%)&uarr;|
> > > > > |----------|----------|----------| ----------|----------|----------|----------|
> > > > > |SeCo| 69.68 | 61.76 | 2.04 | 98.38 | 99.24 | 38.39 |
> > > > > | FPR | 86.07 | 77.80 | 0.67 | 97.88 | 98.80 | 74.91 |
> > > > > | CoinGAN  | 89.41 | 82.30 | 0.47 | 98.56 | 99.63 | 72.14 |
> > > > >
> > > > > The experimental results show that CoinGAN achieves superior performance, surpassing state-of-the-art image-level weakly supervised semantic segmentation models, including FPR and SeCo. Notably, FPR and SeCo were previously the top-performing models across the three datasets, as detailed in Tables 1–3.
> > > > >
> > > > > - **A new application:** Additionally, we have extended our proposal to include a new application: modeling the progression of a specific disease, further showcasing the versatility of CoinGAN as illustrated in our response of **Q3**. The corresponding changes are in  Appendix K. Specifically, we have applied it to model the progression of mediastinal lesions from mild to severe, capturing the gradual worsening of the condition. In this process, varying degrees of intensities are included. Similarly, our proposal is highly adaptable and can be extended to domains such as face recognition, person re-identification, and object tracking, which share similarities with our setting. This generalizability underscores the potential of our work to inspire future research in these broader domains.
> > > > >
> > > > > In summary, the discrepancies within regions of the same image and between different labels of images are the core idea of our work. It is promising in other fields.
> > > > >
> > > > >
> > > > >
> > > > > >**New Q3**. And I still don't know what an "anatomical distribution" is. "Anatomical" suggests a spatial component, but the plots suggest a simple pixel intensity distribution.
> > > > >
> > > > > We have revised the use of "anatomical distribution" in the updated manuscript.

---

### Official Review · Reviewer_dQst · 2024-11-04

**Soundness:** 3
**Presentation:** 2
**Contribution:** 3
**Rating:** 5
**Confidence:** 3

**Summary:**

The paper proposes a weakly supervised semantic segmentation (WSSS) method for lesion segmentation. The main contributions of the paper are two modules: one is called contrastive convolution which focuses on the discrepancies between lesion and healthy structures to reduce the uncertainties in boundaries, the second one is a dual attention mechanism called CSA which learns inter image discrepancy with adversarial training. The experiments are performed on 3 public datasets and the method is compared to generic sota WSSS methods and to methods specific for some medical images, along with the ablation studies. Additionally, the paper demonstrate that WSSS methods that work well on natural images do not perform well on medical datasets. The results show that the proposed method achieves significant improvement.

**Strengths:**

- The idea presented in the paper is interesting.
- The method is validated on sufficiently large datasets and compared with various SoTA methods.
- The results demonstrate that the method achieves remarkable improvement.

**Weaknesses:**

I think the major weakness of the paper is the unclear description and lack of some important details:
- Average buffer and SElayer are crucial components of the proposed architecture; however, the details of these components are not provided in the paper. Please explain how do these components work in detail.
- x_p and x_h used in Figure 2 are not defined in the paper. To my understanding, one of them is the pathological image and the other is the healthy one. However, my understanding brings more questions regarding the datasets used in the experiments. For example, QaTA-Cov19 is a pneumonia benchmark which does not contain healthy images. Where are the healthy images used in this experiment coming from? This question is also valid for the other datasets. Please clarify.
- It is not very clear to me how does the proposed method predict segmentation masks from image-level annotations. As far as I understand, the method converts the pathological images to the healthy ones by removing the pathologies. Are the segmentation masks obtained by taking the difference between the original image and the converted one?

**Questions:**

- How are Average buffer and SElayer components work?
- From which datasets are the healthy images used in the experiments coming from?
- How does the the algorithm predict segmentation masks from image-level annotations? Is it the region obtained after subtracting the input image and its translated version to an healthy image? If so, does this subtraction reveals any false positives? How are they removed, if any?

---

> ### Author Response · Authors · 2024-11-22
> **Response to Reviewer dQst (1/3)**
>
> We sincerely appreciate your valuable feedback, which has been instrumental in improving the methodological aspects of our paper. Below, we address your questions and suggestions point by point.
>
> #### **W1 & Q1.**
> > *Technical details of Average Buffer and SElayer and how they work.*
>
> In our paper, the Average Buffer and SElayer are two essential components of the proposed CSA mechanism. We provide a detailed explanation of both components below.
>
>
> **Average Buffer**
>
> The Average Buffer is a sample buffer designed to store a certain number of reference samples and compute their average representations. These stored samples are dynamically updated with each batch of input samples. The formal mathematical definition is as follows:
>
> Let:
> - $\mathcal{A} = \\{ \mathbf{A}_1, \mathbf{A}_2, \dots, \mathbf{A}_N \\}$ represent the set of $N$ stored reference samples in the Average Buffer, where $N$ is the batch size.
> - $\mathbf{A}_i$ denote the $i$-th sample in the Average Buffer.
>
> The Average Buffer computes the average representation $\mathbf{\bar{A}}$ of the stored samples as:
>
>   $$\mathbf{\bar{A}} = \frac{1}{N} \sum_{i=1}^{N} \mathbf{A}_i$$
>
> where $\mathbf{A} _ i$ refers to the representation derived from the C-Conv module, specifically $\mathbf{C}\cdot \mathbf{F} _ {\text{IFZ}}(\mathbf{X} _ {h})$.
>
> During model training, when a new batch of samples $\\{ \mathbf{A}^{\prime} _ 1, \mathbf{A}^{\prime} _ 2, \dots, \mathbf{A}^{\prime} _ N \\}$ is input to the Average Buffer, the existing samples $\\{ \mathbf{A} _ 1, \mathbf{A} _ 2, \dots, \mathbf{A} _ N \\}$ are dynamically replaced by the new batch $\\{ \mathbf{A}^{\prime} _ 1, \mathbf{A}^{\prime} _ 2, \dots, \mathbf{A}^{\prime} _ N \\}$.
>
> After replacement, the updated Average Buffer is:
>
> $$
> \mathcal{A} _ {new} = \\{ \mathbf{A}^{\prime} _ 1, \mathbf{A}^{\prime} _ 2, \dots, \mathbf{A}^{\prime} _ N \\}
> $$
> and the average representation of the new Average Buffer becomes:
> $$
> \mathbf{\bar{A}} _ {new} = \frac{1}{N} \sum _ {i=1}^{N} \mathbf{A}^{\prime} _ i
> $$
> where $\mathbf{A}^\prime _ i$ is the representation computed by the C-Conv module for the new batch of samples.
>
> The Average Buffer enables CoinGAN to capture rich structural background information from the healthy modality, using the computed average representation as a reference.
>
> **SElayer**
>
> The SElayer is a channel-wise adaptive weighting algorithm originally proposed in [1]. As described in the reference, the SElayer enables the neural network to prioritize the most critical features for the task at hand, boosting the expressive capacity of representations.
>
> >[1] Jie Hu, Li Shen, and Gang Sun (2018), 'Squeeze-and-Excitation Networks,' in Proceedings of the IEEE Conference on Computer Vision and Pattern Recognition, pp. 7132–7141
>
> In CoinGAN, we integrate the SElayer following the Average Buffer component to adaptively reweight the channel-wise average representations $\mathbf{\bar{A}}$, thereby increasing the expressive power of the healthy modality reference representation and improving the utilization of background information in subsequent processing. The detailed algorithm is outlined as follows:
>
>
> **Algorithm 1: SELayer (Squeeze-and-Excitation Layer)**
>
> **Input:** Channel-wise average representations $\mathbf{\bar{A}}$ with shape $(N _ H, N _ W, N _ L)$, where $N _ H$ is the height, $N _ W$ is the width, and $N _ L$ is the number of channels.
>
> **Output**: Recalibrated representations $\mathbf{B}$.
> 1. Compute the channel descriptor $\mathbf{z}$ via global average pooling:
>        $$\mathbf{Z} _ l = \frac{1}{N _ H \times N _ W} \sum _ {i=1}^{N _ H} \sum _ {j=1}^{N _ W} \mathbf{\bar{A}} _ {ijl},  \forall l = 1, 2, \dots, N _ L
>        $$
>
> 2. Pass $\mathbf{Z}$ through a bottleneck architecture comprising two fully connected layers:
>           $$
>           \mathbf{\hat{Z}} = \sigma(\mathbf{W _ 2}(\text{ReLU}(\mathbf{W _ 1} \mathbf{Z} + \mathbf{b _ 1})) + \mathbf{b _ 2})
>           $$
>           where $\sigma$ denotes the sigmoid activation function, and $\mathbf{\hat{Z}}$ is the recalibration vector with shape $(1, 1, N _ L)$.
>
> 3. Recalibrate the input representations $\mathbf{\bar{A}}$ by element-wise scaling with $\mathbf{\hat{Z}}$:
>        $$
>        \mathbf{B} _ {ijl} = \mathbf{\bar{A}} _ {ijl} \cdot \mathbf{\hat{Z}} _ l, \forall i, j, l
>        $$
>
> 4. Return the recalibrated representations $\mathbf{B}$.
>
> This integration of the SElayer refines the CSA mechanism by emphasizing the most relevant features, facilitating the effective exploitation of background information.
>
> To clarify, We will include the updated description of the Average Buffer and SElayer's operation in the manuscript, and supplement the corresponding technical details in the Appendix.

---

> ### Author Response · Authors · 2024-11-22
> **Response to Reviewer dQst (2/3)**
>
> #### **W2 & Q2.**
>
> >*The definitions of $\mathbf{X}_p$ and $\mathbf{X}_h$*
>
> The definitions of $\mathbf{X}_p$ and $\mathbf{X}_h$ are consistent with your interpretation: $\mathbf{X}_p$ denotes the initial representation of the pathological image, as processed by the Embedding Mapping layer in Figure 2, whereas $\mathbf{X}_h$ represents the initial representation of the healthy image, similarly mapped by the Embedding Mapping layer in Figure 2.
>
> >*Where are the healthy images used in this experiment coming from?*
>
> Regarding the source of healthy images, we provide detailed clarifications for the three datasets below:
> - **QaTa-COV19 dataset**: This dataset includes 9,258 COVID-19 chest X-rays with ground truth segmentation masks and 12,544 normal (healthy) chest X-rays as the control group, as described in the README file of QaTa-COV19. Additional details can be found at https://www.kaggle.com/datasets/aysendegerli/qatacov19-dataset.
>
> - **ISIC2018 dataset**: This dataset contains 3,694 skin images with lesions. Inspired by [2], we cropped healthy skin regions from the backgrounds of these images and applied bilinear interpolation, resulting in healthy images (control samples).
>
> - **MoNuSeg dataset:** Similar to the approach in [3], we processed this dataset to derive healthy images by applying morphological operations (image erosion) and Gaussian filtering to obtain the background information.
>
> >[2] Tschandl, Philipp, et al. "Human–computer collaboration for skin cancer recognition." Nature medicine 26.8 (2020): 1229-1234. \
> [3] Yang, Xilin, Bijie Bai, Yijie Zhang, Musa Aydin, Yuzhu Li, Sahan Yoruc Selcuk, Paloma Casteleiro Costa et al. "Virtual birefringence imaging and histological staining of amyloid deposits in label-free tissue using autofluorescence microscopy and deep learning." Nature Communications 15, no. 1 (2024): 7978.
>
> To strengthen the understanding of the methodological and experimental details, we will update the definitions of $\mathbf{X}_p$ and $\mathbf{X}_h$ in the manuscript and further provide the dataset descriptions in the Appendix with the above details.

---

> > ### Author Response · Authors · 2024-11-25
> > **Follow up: the updated manuscript for Reviewer  dQst**
> >
> > Hi, Reviewer dQst, we have revised the manuscript  point by point based on your comments and suggestions, where the corresponding changes have been made (highlighted in blue). The detailed updates are as follows:
> >
> > **W1 & Q1.**
> > - The updated description of the Average Buffer have been revised in line 295 of the updated manuscript and more technical details are updated in Appendix C.
> > - The updated description of the SElayer algorithm have been revised in lines 298-300. More details in this algorithm have been updated in Appendix D.
> >
> > **W2 & Q2.**
> > - The definitions of $\mathbf{X}_p$ and $\mathbf{X}_h$ have been updated in lines 293-294.
> > - The source of healthy images have been updated in Appendix E.
> >
> > **W3 & Q3.**
> > - Our derivation of segmentation masks have been updated in lines 342-343. More details have been revised in Section 3.
> >
> > We hope that the above responses and corresponding changes in the manuscript can address your concerns. If you have any further questions, we are ready and eager to engage in further discussions.

---

> ### Author Response · Authors · 2024-11-22
> **Response to Reviewer dQst (3/3)**
>
> #### **W3 & Q3.**
>
> >How does the proposed method predict segmentation masks from image-level annotations?
>
> Our derivation of segmentation masks aligns with the reviewer's understanding: the segmentation masks were obtained by computing the difference between the original image and the converted one.
>
> First, the C-Conv module learns intra-image representation discrepancies. Specifically, the fine-grained perception differences captured by the Edge Perception Zone (EPZ) convolution and the Internal Feature Zone (IFZ) convolution effectively identify the boundary regions with class changes. This approach mitigates the challenge of ambiguous boundaries in lesion segmentation.
>
> Next, the CSA mechanism employs a Commonality Attention (CA) mechanism to capture similarity representations between the pathological and healthy modalities, and a Specificity Attention (SA) mechanism to capture the discrepancy representations between these modalities. By emphasizing similarity representations and diminishing discrepancy representations, the CSA mechanism facilitates the conversion of pathological representations into healthy ones.
>
> Finally, the standard CycleGAN is used to perform the conversion from the pathological image to the healthy image. This process captures the distribution discrepancy between modalities, and the difference between the original image and the converted healthy image serves as the segmentation mask for the lesion region.
>
>
> >Does this subtraction reveals any false positives? How are they removed, if any?
>
> We believe this subtraction will not result in false positives. Specifically, as stated above, the difference between the original image and the converted one constitutes the final segmentation mask. In this process, false positives primarily arise from inconsistencies in the background structure between the converted image and the original one, which may lead the model to misclassify background information as lesions.
>
>
> Our proposed CoinGAN employs two mechanisms to mitigate false positives:
> - The **commonality attention (CA)** mechanism within the CSA mechanism captures background structures tailored to the target object. This prevents the background representation in the converted image from deviating from that of the original image.
>
> - The **identity mapping loss** $\mathcal{L} _ {identity}$, inherently integrated into CycleGAN, ensures consistency between the converted image and the original image, as defined by:
> $$
> \mathcal{L} _ {identity}= ||G(\mathbf{F} _ {\text{CSA}}(\mathbf{C} \cdot \mathbf{F} _ {\text{IFZ}}(\mathbf{X}_p)))-\mathbf{X} _ p||
> $$
>
> where $\mathbf{C} \cdot \mathbf{F} _ {\text{IFZ}}(\cdot)$ is the C-Conv module, $\mathbf{F} _ {\text{CSA}}(\cdot)$ is the CSA mechanism. $G(\cdot)$ is the genrator in CycleGAN. $\mathbf{C}$ is the category activation maps from the C-Conv module and $\mathbf{F} _ {\text{IFZ}}(\cdot)$ is the Internal Feature Zone (IFZ) convolution. These mechanisms ensure the consistency of background information, hence reducing the risk of false positives.

---

> > ### Comment · Reviewer_dQst · 2024-11-28
> > **Further clarification needed about mitigating false positives**
> >
> > Thanks to the authors for the detailed response. I need more clarification about the false positives. In the above response, it is mentioned that this subtraction will not result in false positives and the false positives primarily arise from the inconsistencies in the background structure. I didn't really get why can't there be any false positives in the foreground region. To my understanding, an anomaly image is converted to its healthy version and the anomalies are detected from the difference of these images. However, the network can add additonal anomalies, bluriness and so on to the healthy part of the input image. How does the method ensure that such artefacts are not created in test time? I understand that the loss terms and the proposed components help; however, I would be very surprised if the network remain the healthy part of the input image untouched and only modifies the anomaly part. I don't remember a specific example discussing this issue with GANs; however, this is very common in unsupervised anomaly detection with VAEs, e.g. in [1].
> >
> > [1] Chen et al. "Unsupervised Detection of Lesions in Brain MRI using constrained adversarial auto-encoders", https://arxiv.org/pdf/1806.04972

---

> ### Author Response · Authors · 2024-11-29
> **Further clarification for false positives**
>
> Thank you for your response and for suggesting the relevant reference [1]. It provides a good discussion about false positives.
>
> We agree that the deep neural network may introduce potential anomalies, bluriness and so on to the healthy part of the input image, which could potentially lead to false positives. The mentioned reference work [1] introduced a regularization loss term to mitigate this issue, where this newly added loss term imposed consistency in the latent representations between the original and reconstructed representations. Similarly, our CoinGAN adopted a similar strategy by incorporating the identity mapping loss, inherently integrated into CycleGAN, to keep the consistency between the converted image and the original image. This helps reduce the risk of false positives by ensuring that the converted image closely matches the original in its structural and contextual features. Additionally, the proposed CSA mechanism further alleviates these false positives by capturing background structures tailored to the target object, preventing the converted image from deviating from that of the original image, as outlined in our response in **W3 & Q3**.
>
>
> Finally, we acknowledge that false positives are difficult to completely eliminate owing to the inherent complexity of deep learning and the intricacies of medical data, but our work has significantly reduced the risk of false positives.
> Besides, It is worth noting that false positives would be inevitable in most deep learning-based methods. It remains a critical area for future exploration.

---

### Author Response · Authors · 2024-11-22
**Thank you for your reviews**

We would like to extend our profound gratitude to all the reviewers for their insightful comments and constructive suggestions, which have played a pivotal role in clarifying our contributions and further improving the quality of our paper.

In response to the reviews, we have conducted the following additional experiments during the rebuttal stage:
- Evaluation of CoinGAN on a different medical imaging modality and less common diseases (Reviewer SjzH (W1 & Q1)).
- Inclusion of a robustness study to demonstrate the robustness of our proposed CoinGAN to inaccuracies and variability in image-level labels (Reviewer SjzH (Q3)).
- Comparison with a state-of-the-art domain-specific diffusion-based model (Reviewer EB1C (W2)).
- Exploration of the progression of mediastinal lesions with labels reflecting severity levels rather than binary presence/absence (Reviewer LD6Q (Q3)).


Furthermore, we have provided responses to each reviewer's comments individually and in detail below, which will also be reflected in the revised manuscript. If there are any additional questions or comments, we stand ready and eager to engage in further discussions.

We are currently revising the paper and will submit the updated version shortly.

---

### Meta-Review · Area_Chair_sWgz · 2024-12-20

**Metareview:**

The paper proposes a weakly-supervised semantic segmentation framework that combines contrastive convolution and dual attention mechanisms (CSA), achieving state-of-the-art performance on medical imaging datasets. However, reviewers raised concerns about the clarity, originality, and generalizability of the method. The ideas presented are not novel, as they build on existing work, and the paper lacks adequate comparisons with recent GAN- and diffusion-based techniques. Overall, the paper was considered to make incremental contributions and suffers from insufficient clarity, requiring significant revisions before being ready for publication. As a result, the consensus was to reject the paper.

**Additional Comments On Reviewer Discussion:**

While the method addresses specific challenges in medical imaging, the generalization to other domains remain unconvincing.

---

### Decision · Program_Chairs · 2025-01-22

Reject